

# RNA sequencing-based exploration of the effects of far-red light on lncRNAs involved in the shade-avoidance response of *D. officinale*

Hansheng Li[1], Wei Ye[2], Yaqian Wang[1], Xiaohui Chen[3], Yan Fang[1] and Gang Sun[1]

[1] College of Resources and Chemical Engineering, Sanming University, Sanming, China
[2] The Institute of Medicinal Plant, Sanming Academy of Agricultural Science, Shaxian, China
[3] Institute of Horticultural Biotechnology, Fujian Agriculture and Forestry University, Fuzhou, China

## ABSTRACT

*Dendrobium officinale* (*D. officinale*) is a valuable medicinal plant with a low natural survival rate, and its shade-avoidance response to far-red light is as an important strategy used by the plant to improve its production efficiency. However, the lncRNAs that play roles in the shade-avoidance response of *D. officinale* have not yet been investigated. This study found that an appropriate proportion of far-red light can have several effects, including increasing the leaf area and accelerating stem elongation, in *D. officinale*. The effects of different far-red light treatments on *D. officinale* were analysed by RNA sequencing technology, and a total of 69 and 78 lncRNAs were differentially expressed in experimental group 1 (FR1) versus the control group (CK) (FR1-CK) and in experimental group 4 (FR4) versus the CK (FR4-CK), respectively. According to GO and KEGG analyses, most of the differentially expressed lncRNA targets are involved in the membrane, some metabolic pathways, hormone signal transduction, and O-methyltransferase activity, among other functions. Physiological and biochemical analyses showed that far-red light promoted the accumulation of flavonoids, alkaloids, carotenoids and polysaccharides in *D. officinale*. The effect of far-red light on *D. officinale* might be closely related to the cell membrane and $Ca^{2+}$ transduction. Based on a Cytoscape analysis and previous research, this study also found that MSTRG.38867.1, MSTRG.69319.1, and MSTRG.66273.1, among other components, might participate in the far-red light signalling network through their targets and thus regulate the shade-avoidance response of *D. officinale*. These findings will provide new insights into the shade-avoidance response of *D. officinale*.

# INTRODUCTION

*Dendrobium officinale* Kimura et Migo (*D. officinale*) is a valuable medicinal plant species that can be used to nourish the stomach and strengthen the body's immunity and that exhibits antitumour and antiaging activity (*Sun et al., 2017*). Modern chemical and

Corresponding author
Gang Sun, sungang@nenu.edu.cn

pharmacological studies have shown that *D. officinale* is rich in polysaccharides, alkaloids, phenols and other secondary metabolites (*Sun et al., 2017*). Researchers have used different methods to increase the yield of the pharmaceutical ingredients of *D. officinale*. For example, *Wang et al. (2016)* compared the medicinal value of *D. officinale* under different cultivation practices and found that the content of polysaccharides, flavonoids and polyphenols was the highest under shaded conditions. *Qin et al. (2018)* determined the polysaccharide content of *D. officinale* during different years of growth, and their results showed that 5-year-old *D. officinale* plants had the highest polysaccharide content, followed by 3-year-old plants. *Dai et al. (2018)* showed that plant growth regulators can promote the growth and development of *D. officinale* plants, increase their resistance to oxidative enzymes, and regulate their chlorophyll content to increase their stress resistance and the accumulation of polysaccharides.

During plant growth, various internal and external factors can affect the synthesis of medicinal components, and light regulation is considered one of the important methods for improving the yield of these components. A study of the accumulation of polysaccharides in the original bulbs of *D. officinale* under different LED illumination conditions revealed that red-blue mixed light exerts the most obvious effect on the accumulation of polysaccharides. The yield of polysaccharides under 1:3 red light:blue light was the most ambiguous (*Hou et al., 2013*). *Xu, Cui & Guo (2012)* reported that the polysaccharide content of *D. officinale* protocorms was highest (up to 16.88%) after 30 d of exposure to a light intensity of 2000 1×. However, the optimal alkaloid content (up to 0.028%) was detected after 30 d of exposure to a light intensity of 500 lx.

One of the most important effects of 730-nm far-red light on plants is the shade-avoidance response (*Chen & Semiconductors, 2015*). The shade-avoidance response of most plants involves a unique set of morphological changes and characteristics, such as an increase in the leaf area, plant height, and dry matter accumulation and changes in the photosynthetic physiological characteristics (*Yang et al., 2017*; *Schambow et al., 2019*). *Kasulin, Agrofoglio & Botto (2013)* reported that the height of *Arabidopsis thaliana* (*A. thaliana*) can increase through hypocotyl and stem elongation so that the plants protrude higher than others in a population. The red light:far-red light (R:FR) ratio in the upper layer of a population is similar to the normal value, and plants in this layer receive relatively more available light energy. The shade-avoidance response of plants can also regulate the growth period of the plant such that the flowering period occurs sooner, the vegetative period is shortened, and reproductive growth is completed more quickly (*Kutschera & Briggs, 2013*). Therefore, 730-nm far-red light might promote an increase in the leaf area and the elongation of stem segments, among other effects, in *D. officinale*, and these effects would in turn improve the production efficiency.

The understanding of the molecular mechanism through which far-red light affects the plant shade-avoidance response has increased in model plant species. The plant shade-avoidance response is mediated mainly by phytochrome B (PHYB), which regulates the expression of downstream genes by phytochrome-interacting bHLH factors (PIFs) (*Amanda, Herrera & Maloof, 2016*). Phytochrome is transported into the nucleus after being converted into an active far-red light-type (Pfr) configuration by light, and Pfr in

the nucleus can directly interact with PIFs (*Amanda, Herrera & Maloof, 2016*). PIFs are phosphorylated in comparison with photoactivated phytochrome and are then degraded by the 26S proteasome. The stability of PIFs plays an important role in the plant shade-avoidance response (*Xie et al., 2017*). PIF4 and PIF5 mediate stalk elongation in plants. Under a low R:FR ratio, Pfr is converted into the inactive red light-type (Pr) configuration by light and is exported from the nucleus, which enhances the stability of PIF4 and PIF5 and promotes regulation of the expression of genes involved in stem elongation (*Lorrain et al., 2008*; *Shi et al., 2018*). In addition, cryptochrome can mediate the shade-avoidance response through the CRY-SPA1/COP1 pathway and inhibit a plant's shade-avoidance response by preventing the degradation of the positive photoregulatory regulators elongated hypocotyl 5 (HY5) and long hypocotyl in far red 1 (HFR1) (*Robson et al., 2010*; *Sellaro, Yanovsky & Casal, 2011*). Recent studies have shown that SPA1 is one of three negative regulators of photomorphogenesis and might participate in the plant shade-avoidance response by regulating the accumulation of phytochrome and cryptochrome (*Sellaro, Yanovsky & Casal, 2011*).

The involvement of long noncoding RNAs (lncRNAs) in the plant shade-avoidance response has not been previously studied. Due to the rapid development of sequencing technology, low-abundance transcripts at the genome-wide transcription level have been detected. lncRNAs, which are noncoding RNAs longer than 200 nucleotides, have extremely complex and important biological functions; they not only regulate gene expression at the epigenetic, transcription, and posttranscriptional levels but also participate in the regulation of many various biological processes, such as genomic imprinting, chromosome remodelling, and transcriptional activation (*Garima et al., 2017*; *Wang et al., 2017*). Recent studies have shown that lncRNAs play an important role in the plant response to external factors. The treatment of *A. thaliana* plants with drought, cold and high salt stress significantly changed the expression levels of 1832 lncRNAs. The expression of individual lncRNAs was even upregulated by 22-fold, which indicated that lncRNAs are important regulators of plant responses to external factors and thereby help plants better adapt to different environmental conditions (*Liu et al., 2012*). *Wang et al. (2014)* identified a large number of lncRNAs that are activated in response to light in *A. thaliana*, and *Shuai et al. (2014)* reported 504 lncRNAs that are activated in response to drought in *Populus euphratica* (*P. euphratica*). Studies on the response of lncRNAs to external factors have been performed in various plant species, such as *A. thaliana*, *Oryza sativa* (*O. sativa*), *P. euphratica*, *Triticum aestivum*, *Zea mays* and *Lycopersicon esculentum*, but the effects of far-red light on lncRNAs have not yet been investigated in *D. officinale*.

In this study, we used high-throughput sequencing technology to identify putative lncRNAs and investigated their expression profiles in *Dendrobium* under different illumination patterns. The specific lncRNAs identified by comparing and analysing the sequence data from the treatment and control groups were further studied to assess their involvement in the shade-avoidance response of *D. officinale* induced by far-red light. Moreover, the signal transduction pathway of the lncRNAs involved in the shade-avoidance response of *D. officinale* was determined. These findings provide new insights for the high-yield production of medicinal components of *D. officinale*.

## MATERIALS AND METHODS

### Plant material and light treatments

Tissue culture-generated seedlings of *D. officinale* were provided by the Sanming Academy of Agricultural Sciences, and tissue culture-generated seedlings with three to four true leaves, a seedling height of approximately 25 mm, a stem diameter of approximately 1.8 mm and a leaf area of approximately 30 mm$^2$ were selected for light treatment. The selected tissue culture-generated *D. officinale* seedlings were placed in a growth chamber for 120 d, and 15 bottles were used for each treatment. The light treatments comprised exposure to red light (660 nm), blue light (450 nm), and far-red light (730 nm); in all treatments, the total light intensity was 200 $\mu$mol m$^{-2}$ s$^{-1}$, the photoperiod was 12 h d$^{-1}$, the humidity ranged from 55% to 60%, and the temperature was 25 $\pm$ 2 °C. In this study, a group treated with a red light intensity:blue light intensity:far-red light intensity ratio of 100:100:0 served as the control group (CK), and the groups treated with red light intensity:blue light intensity:far-red light intensity ratios of 80:80:40 (experimental group 1 (FR1)), 60:60:80 (experimental group 2 (FR2)), 50:50:100 (experimental group 3 (FR3)), 40:40:120 (experimental group 4 (FR4)), and 20:20:160 (experimental group 5 (FR5)) served as the experimental groups. The tissue culture-generated seedlings of *D. officinale* were maintained on half-strength Murashige and Skoog (MS) media (6 g L$^{-1}$ agar and 30 g L$^{-1}$ sucrose, pH 5.8) supplemented with 1 g L$^{-1}$ activated carbon and 50 g L$^{-1}$ mashed banana and subcultured every 120 d. Half-strength Murashige and Skoog (MS) media (6 g L$^{-1}$ agar and 30 g L$^{-1}$ sucrose, pH 5.8) supplemented with 1 g L$^{-1}$ activated carbon was used for the light treatments. The all samples were frozen in liquid nitrogen and stored at −80 °C for high-throughput sequencing, nucleic acid extraction, functional metabolite content determination and antioxidant enzyme activity assays, etc.. The FR1 versus CK, FR4 versus CK, and FR4 versus FR1 comparisons are denoted FR1-CK, FR4-CK, and FR4-FR1, respectively.

### lncRNA library construction and Illumina HiSeq sequencing

According to their phenotypic differences, samples of *D. officinale* after three treatments—CK, FR1 and FR4—were selected for high-throughput sequencing, and each treatment involved three biological replicates. Total RNA extraction and determination (Fig. S1; Table S1) referred the method described in *Li et al. (2019)*. Ribosomal RNA (rRNA) was eliminated from the purified RNA using the Ribo-Zero rRNA Removal Kit (Epicentre, Madison, WI, USA). Strand-specific cDNA was synthesized to construct nine *D. officinale* sequencing libraries using the TruSeq® Stranded Kit (Illumina), ribonuclease H, and DNA polymerase I. Raw data in the fastq format were obtained using the Illumina HiSeq sequencing platform, and low-quality reads, reads with a high N content, and adapter contamination were removed to obtain clean reads. Sequencing library preparation and high-throughput sequencing were subsequently performed using the Illumina HiSeq platform (Beijing, China). All sequencing data of *D. officinale* after the different light treatments were deposited in the National Center for Biotechnology Information (NCBI) Sequence Read Archive (accession number PRJNA638348).

The clean reads were aligned to the *Dendrobium officinale* reference genome (*Zhang et al., 2016*) using Hierarchical Indexing for Spliced Alignment of Transcripts (HISAT) software for transcript assembly, and transcripts with lengths below 200 bp and lower expression (FPKM ≤ 0.5) were filtered out. The reference genome version of *Dendrobium officinale* in this manuscript was updated on April 11, 2019. This new version can be found on the website (https://www.ncbi.nlm.nih.gov/genome/?term=txid906689[orgn]). The transcripts were identified as lncRNAs or mRNAs using the CPAT, CPC, CNCI and Pfam protein domain databases (*Chen et al., 2018b*; *Chen et al., 2018a*). The clean reads were mapped to the *Dendrobium officinale* reference genome to quantify the gene comparison rate using Bowtie v2.2.3 software (*Langmead, 2012*). The expression of transcripts, lncRNAs and mRNAs was calculated using RSEM v1.2.12 software (*Li, 2011*). Differential expression analysis using the DESeq R package (1.10.1) (*Yu et al., 2012*). The *P*-values were adjusted using Benjamini's approach (*Benjamini & Hochberg, 1995*). Genes with an adjusted *P*-value <0.01 and a $|\log_2(\text{fold change})|>1$, as determined with DESeq, were defined as differentially expressed genes.

## lncRNA target prediction and annotation

The cis roles of lncRNAs involve their action on neighbouring target genes. We searched coding genes that were 10 kb/100 kb upstream/downstream of lncRNAs and subsequently analysed their function. To understand the functions of lncRNAs and their targets, Non-Redundant Protein Sequence Database (NR), Nucleotide Sequence Database (NT), clusters of euKaryotic Orthologous Groups (KOG), Kyoto Encyclopedia of Genes and Genomes (KEGG), and Swiss-Prot annotations were assigned to the novel and known assembled mRNAs using Diamond (*Buchfink, Xie & Huson, 2015*) or Blast (*Lobo, 2012*). The gene ontology (GO) and InterPro annotations were assigned using Blast2GO (*Conesa et al., 2005*) and InterProScan5 (*Quevillon et al., 2005*), respectively. Cytoscape_v3.6.0 software was used for the network mapping of the lncRNAs and their targets.

## Secondary metabolite determination

Freeze-dried grains of *D. officinale* stems and leaves were extracted as previously described (*Li et al., 2018*). The absorbance was measured with a UV-visible spectrophotometer (Evolution 350, Thermo Fisher, MA, USA), and the wavelengths of polysaccharides, flavonoids and alkaloids were measured at 485 nm, 510 nm and 416 nm, respectively (*Li et al., 2018*). The polysaccharide, flavonoid and alkaloid contents in the *D. officinale* stems and leaves were calculated according to established standard curves.

The carotenoids were determined using the method described by *Fan et al. (2017)*. Two grams of fresh *D. officinale* stems and leaves was weighed and added to 10 mL of acetone, and the mixture was then incubated in the dark until the *D. officinale* material turned white. The absorbance of the sample at 475 nm was measured using a UV-visible spectrophotometer, and the carotenoid contents were calculated as described by *Fan et al. (2017)*.

## Determination of physiological and biochemical indicators

Determination of superoxide dismutase (SOD) activity. *D. officinale* leaves (fresh weight) were ground in liquid nitrogen. First, 1 g of powder was weighed and dissolved in 10 mL of extract, and the homogenate of *D. officinale* was centrifuged at 8000× g and 4 °C for 15 min. The supernatant was then removed, placed on ice and collected for determination of SOD activity. The activity of SOD in *D. officinale* leaves was detected using commercial kits (Keming, Suzhou, China) according to the manufacturer's instructions.

Determination of the relative membrane permeability. The relative conductivity of the leaves of *D. officinale* was measured using a conductivity metre (MIK-EC8.0, Meacon, Hanzhou, China). Ten pieces of 1-cm-wide leaves were cut and placed in a beaker, and 40 mL of deionized water was then added. The conductivity was measured immediately after the addition of deionized water ($P_0$) and again 10 min later ($P_1$). The beaker was then boiled in a boiling water bath for 10 min and cooled to room temperature; subsequently, deionized water was added to the marked height, and the conductivity ($P_2$) was measured. The ratio of the conductivity of the extract before and after boiling indicates the permeability of the cell membrane: relative membrane permeability (%) = 100 ($P_1$-$P_0$)/($P_2$-$P_0$).

Determination of the calmodulin (CaM) content. The CaM samples were prepared according to *Wang et al. (2010a)* and *Wang et al. (2010b)*. Two grams of *D. officinale* leaves after the different light treatments was weighed and ground with liquid nitrogen to obtain a homogenate. Prechilled cell lysate [50 mmol $L^{-1}$ Tris–HCl pH 7.4, 50 mmol $L^{-1}$ NaCl, two mmol L $L^{-1}$ EDTA, one mmol $L^{-1}$ PMSF, one mmol $L^{-1}$ $\beta$-mercaptoethanol, and 1% (W/W) Triton X-100] was added at a ratio of 1:2 (W/V). The cells were disrupted using ultrasonic washing equipment (KQ-200SPDE, Keqiao, Guangdong, China), boiled at 95 °C for 5 min, and centrifuged at 10,000 × g and 4 °C for 20 min. After discarding the precipitate, the supernatant was the crude extract containing CaM, and the content of CaM was determined by ELISA. According to the operating instructions provided with the plant calmodulin ELISA detection kit (Keming, Suzhou, China), the absorbance values of the standard series solutions and sample solutions were measured with a microplate reader (Multiskan Sky, Thermo Fisher Scientific, MA, USA) at a wavelength of 450 nm, and the CaM content of *D. officinale* leaves was calculated according to the established standard curve.

## Quantitative real-time PCR (qRT-PCR) analysis

Total RNA from *D. officinale* leaves after the different light treatments was used for qPCR validation of lncRNAs and mRNAs. Total RNA extraction and determination referred the method described in *Li et al. (2019)*. The cDNA synthesis, reaction system and procedures, etc. referred the method described by *Lin & Lai (2010)* and *Lin & Lai (2013)*. The qPCR assays were performed under standard conditions for 40 cycles on a LightCycler 480 Real-Time PCR System (Roche, Switzerland). Different lncRNA and target genes use different annealing temperatures. A five-fold dilution gradient of the mixed sample was used to obtain the standard curve and determine the reliability of the standard curve (Figs.S2–S3). The *ACTIN* gene was used as reference gene and the relative lncRNA and

**Table 1  Phenotype of *D. officinale* after different light treatments.**

| Treatment | CK | | | FR1 | | | FR4 | | |
|---|---|---|---|---|---|---|---|---|---|
| | Average value | Standard deviation | Duncan (5%) | Average value | Standard deviation | Duncan (5%) | Average value | Standard deviation | Duncan (5%) |
| Plant height (mm) | 50.90 | 0.64 | c | 58.20 | 0.87 | b | 68.22 | 0.64 | a |
| Stem diameter (mm) | 4.45 | 0.07 | b | 4.62 | 0.05 | a | 4.64 | 0.04 | a |
| Leaf area (mm²) | 93.34 | 1.09 | c | 105.00 | 1.24 | b | 125.37 | 1.19 | a |
| Number of leaves | 5 | – | – | 5 | – | – | 5 | – | – |

mRNA expression levels were calculated with the $2^{-\Delta\Delta Ct}$ method. The primers used in this assay are listed in Table S2.

## Statistical analysis

The quantitative results concerning the gene expression, physiological and biochemical indexes and functional metabolites of *D. officinale* were obtained from at least three biological replicates. The effects of the different light treatments on *D. officinale* were analysed by one-way analysis of variance (ANOVA) followed by Duncan's test using SPSS version 19.0 (IBM, NY, USA). The figures were generated with GraphPad Prism 6.0 software (GraphPad Software, CA, USA) and Omic-Share online software (Biomarker Technologies, Beijing, China).

## RESULTS

### Growth state of *D. officinale* under different light treatments

Far-red light can affect the shade-avoidance response of *D. officinale*, and these effects manifest as a series of unique morphological changes. The largest leaf area of *D. officinale* was obtained with the FR4 treatment (125.37 mm²), followed by the FR1 (105.00 mm²) and CK treatments (93.34 mm²) (Table 1; Figs. 1A–1F). The highest plant height was also obtained with the FR4 (68.22 mm), followed by the FR1 treatment (58.20 mm), whereas the lowest plant height was obtained with the CK treatment (50.90 mm) (Table 1; Figs. 1A–1F). The stems of the plants under the FR4 and FR1 treatments were slightly thicker than those obtained with the CK treatment (Table 1; Figs. 1A–1F). Therefore, the results showed that far-red light exerts a significant effect on the height, stem segments and leaf area of *D. officinale* and that the FR4 treatment was the most conducive treatment to increasing the leaf area and accelerating stem elongation in *D. officinale*.

### Sequencing and assembly of transcriptomic data

This study revealed that an appropriate proportion of far-red light affected the shade-avoidance response of *D. officinale* by increasing the leaf area and accelerating stem elongation, among other effects, and might thus increase the production of *D. officinale*. To investigate the lncRNAs related to the effects of far-red light on the shade-avoidance response of *D. officinale*, three biological replicate samples after the CK, FR1, and FR4 treatments were subjected to transcriptome sequencing analysis using the Illumina HiSeq

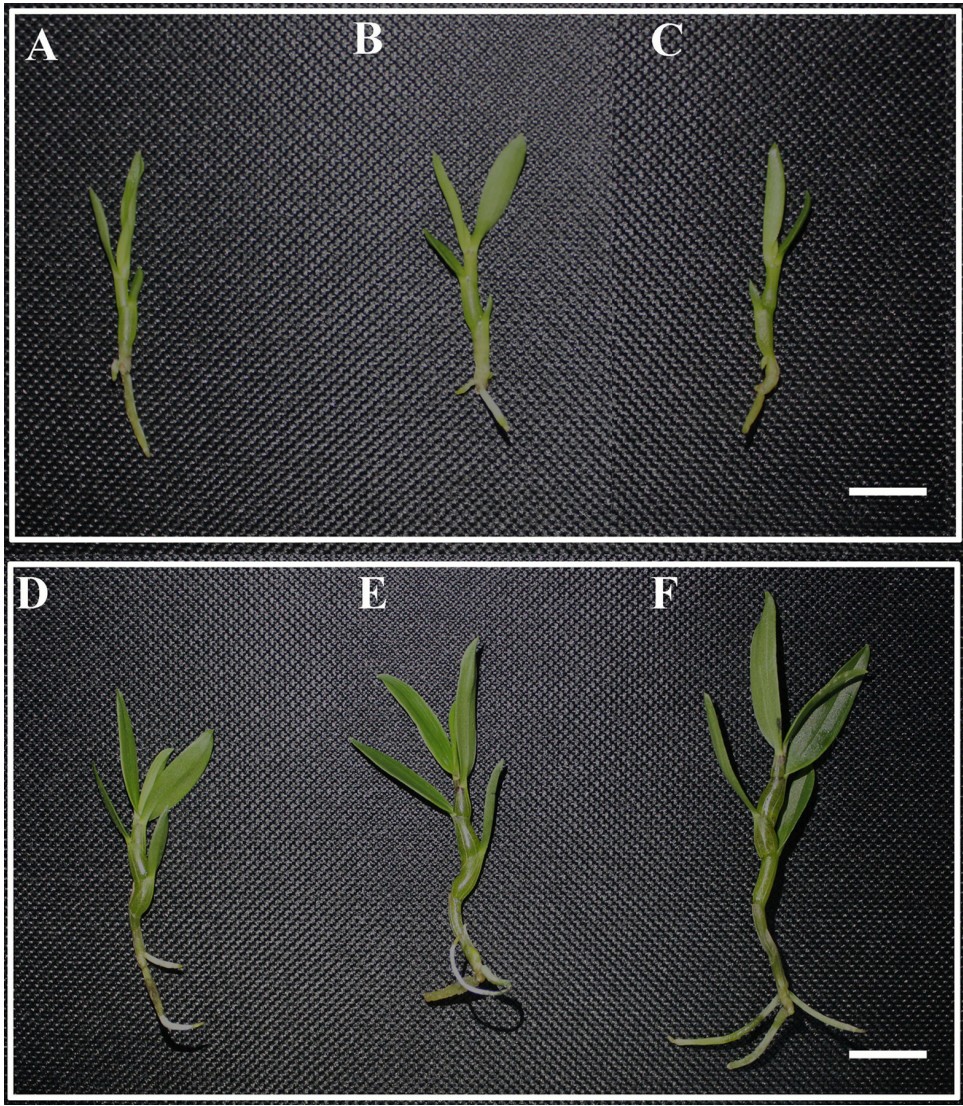

**Figure 1** **Phenotype of *D. officinale* under different light treatments.** (A–C) Phenotype of *D. officinale* before the different light treatments; (D–E) Phenotype of *D. officinale* after the different light treatments. A and D, CK; B and E, FR1; C and F, FR4. Red light intensity:blue light intensity:far-red light intensity ratios: 100:100:0, CK; 80:80:40, FR1; and 40:40:120, FR4. Bars = 10 mm.

platform. After the removal rRNAs and adapter sequences, among other contaminations, an average of 9.66 Gb of clean data were obtained for each treatment. The RNA-seq data of *D. officinale* after the different light treatments yielded 65,759,200 to 88,743,106 reads (Table 2). The mapping ratio of *D. officinale* reference genomes in all treatments approximately 84.16%–86.69%, and uniquely mapping ratio approximately 78.40%–80.01% (Table 2). The sequencing data indicated that the sequencing reads could better match the *D. officinale* reference genome. In addition, the Q20 and Q30 values of the *D. officinale* samples were higher than 95.47% (Table 2), demonstrating the high reliability of the *D. officinale* sequencing data.

**Table 2 mRNA and lncRNA results from nine _D. officinale_ libraries.**

| Samples | Total reads | Total mapped reads (%) | Uniquely mapped reads (%) | Q20 (%) | Q30 (%) | GC content (%) |
|---|---|---|---|---|---|---|
| CK1 | 88,743,106 | 85.91 | 78.90 | 98.73 | 95.69 | 42.48 |
| CK2 | 66,153,614 | 86.69 | 80.01 | 98.85 | 96.01 | 43.11 |
| CK3 | 74,195,304 | 86.19 | 78.87 | 98.87 | 95.95 | 42.46 |
| FR1-1 | 65,759,200 | 86.51 | 79.94 | 98.71 | 95.79 | 42.66 |
| FR1-2 | 66,959,230 | 85.48 | 79.32 | 98.61 | 95.49 | 42.52 |
| FR1-3 | 77,913,256 | 85.47 | 78.40 | 98.53 | 95.47 | 43.02 |
| FR4-1 | 66,569,014 | 85.03 | 79.12 | 98.78 | 95.76 | 43.16 |
| FR4-2 | 67,769,478 | 85.26 | 79.58 | 98.74 | 95.66 | 43.00 |
| FR4-3 | 70,015,212 | 84.16 | 79.01 | 98.66 | 95.51 | 42.65 |

## Analysis of differentially expressed lncRNAs under different light treatments

The results from the transcriptome sequencing yielded 3,086 new genes, 2,125 differentially expressed genes, 3,770 lncRNAs, and 136 differentially expressed lncRNAs (DE lncRNAs) (Tables S3–S11). To study the expression of lncRNAs in _D. officinale_, all lncRNAs were divided into three categories according to their expression levels: high expression (fragments per kilobase of transcript per million mapped reads (FPKM) >50), moderate expression (5 ≤ FPKM ≤ 50) and low expression (FPKM < 5). The results showed that the expression levels of most genes were downregulated, whereas those of a few genes were upregulated (Fig. 2A). All lncRNAs were analysed, and the DE lncRNAs were identified. The results showed that lncRNAs could be regulated by the different light treatments. The FR1-CK comparison identified a total of 69 DE lncRNAs, including 30 upregulated and 39 downregulated lncRNAs, and in the FR4-CK comparison, 78 DE lncRNAs, including 49 upregulated and 29 downregulated lncRNAs, were identified. Among these lncRNAs, 11—seven upregulated and four downregulated lncRNAs—were found in both the FR1-CK and FR4-CK comparisons (Fig. 2B). A cluster analysis revealed that the 136 DE lncRNAs identified after the different light treatments could be divided into six expression patterns (Fig. 2C).

## GO analysis of DE lncRNAs in _D. officinale_

To better understand the changes in the transcriptome of _D. officinale_ under far-red light treatment, a GO analysis was performed using the results from the three different comparisons: FR1-CK, FR4-CK, and FR4-FR1 (Table 3).

Based on the results from the FR1-CK comparison, a GO analysis of biological processes revealed that the most relevant biological processes of the DE lncRNA target genes included leaf shaping, maturation of 5.8S rRNA, response to brassinosteroids, the ubiquinone-6 biosynthesis process, and assembly of the large subunit precursor of preribosomes. The most relevant biological processes identified based on the results from the FR4-CK comparison included the ubiquinone-6 biosynthesis process and sucrose biosynthesis process, whereas

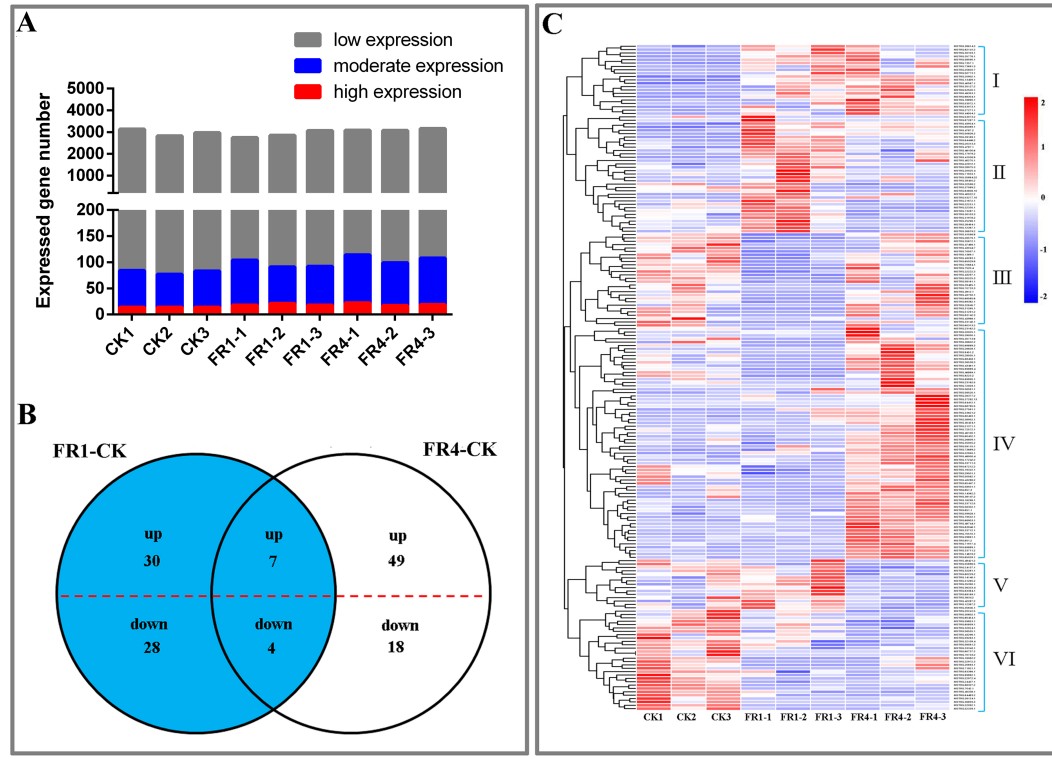

**Figure 2** **DE lncRNAs under the different light treatments.** (A) Number of lncRNAs with high (FPKM > 50), moderate (5 ≤ FPKM ≤ 50) and low (FPKM < 5) expression in each library. (B) Venn diagram showing the numbers of unique and commonly regulated lncRNAs identified from the FR1-CK and FR4-CK comparisons. (C) DE lncRNAs in response to different light treatments. The scale from red to blue corresponds to the numerical value of $\log_2$ (FPKM) from high to low.

the analysis of the FR4-FR1 comparison revealed that the most relevant biological process was leaf shaping.

The most enriched cell components obtained from the analysis of the FR1-CK comparison included signal recognition particle endoplasmic reticulum targeting and the HAUS complex, whereas the analyses of the FR4-CK and FR4-FR1 comparisons showed that the most relevant cell components were the membrane and cis-Golgi networks, respectively.

Based on the results from the FR1-CK comparison, a GO analysis of molecular functions revealed that the most relevant molecular function of the DE lncRNA target genes was translation initiation factor activity. In contrast, the most relevant molecular functions obtained from the analysis of the FR4-CK comparison included O-methyltransferase activity and ureidoglycolate hydrolase activity, and that found from the analysis of the FR4-FR1 comparison was channel activity.

In summary, the shade-avoidance response of *D. officinale* under far-red light might be significantly related to leaf shaping, brassinosteroid synthesis, the ubiquinone-6 biosynthesis process, membranes, and methylation, among other processes and components.

Li et al. (2021), *PeerJ*, DOI 10.7717/peerj.10769

**Table 3** Top five GO terms obtained from an enrichment analysis of the target genes of DE lncRNAs in *D. officinale* under different light treatments.

| | ID | Description | q value | | | geneID |
|---|---|---|---|---|---|---|
| | | | FR1-CK | FR4-CK | FR4-FR1 | |
| **Biological process** | GO:0010358 | leaf shaping | 0.01280 | | 0.02798 | gene-MA16_Dca015315;gene-MA16_Dca018761 |
| | GO:0000460 | maturation of 5.8S rRNA | 0.01280 | | | gene-MA16_Dca018926;gene-MA16_Dca018927 |
| | GO:0009741 | response to brassi-nosteroid | 0.01280 | | 0.03869 | gene-MA16_Dca015315;gene-MA16_Dca018761 |
| | GO:1901006 | ubiquinone-6 biosynthetic process | 0.01280 | 0.07223 | | gene-MA16_Dca014146;gene-MA16_Dca014147 |
| | GO:1902626 | assembly of large subunit precursor of preribosome | 0.01280 | | | gene-MA16_Dca018926;gene-MA16_Dca018927 |
| | GO:0097503 | sialylation | | 0.07223 | | gene-MA16_Dca009824;gene-MA16_Dca013452 |
| | GO:0005986 | sucrose biosynthetic process | | 0.08464 | | gene-MA16_Dca009826 |
| | GO:0006296 | nucleotide-excision repair, DNA inci-sion, 5′ to lesion | | 0.08464 | | gene-MA16_Dca007683 |
| | GO:0006428 | isoleucyl-tRNA aminoacylation | | 0.08464 | | gene-MA16_Dca009286 |
| | GO:0016311 | dephosphorylation | | | 0.03869 | gene-MA16_Dca000617;gene-MA16_Dca013817;gene-MA16_Dca015948; gene-MA16_Dca016094;gene-MA16_Dca024052;gene-MA16_Dca025271 |
| | GO:0070588 | calcium ion trans-membrane trans-port | | | 0.03869 | gene-MA16_Dca015955;gene-MA16_Dca022336;gene-MA16_Dca022337 |
| | GO:0016132 | brassinosteroid biosynthetic process | | | 0.03869 | gene-MA16_Dca015315;gene-MA16_Dca018760;gene-MA16_Dca018761 |

Li et al. (2021), *PeerJ*, DOI 10.7717/peerj.10769

**Table 3** (*continued*)

| | ID | Description | q value | | | geneID |
|---|---|---|---|---|---|---|
| | | | FR1-CK | FR4-CK | FR4-FR1 | |
| **Cellular component** | GO:0005786 | signal recognition particle, endoplasmic reticulum targeting | 0.10451 | | | gene-MA16_Dca000647;gene-MA16_Dca000982 |
| | GO:0070652 | HAUS complex | 0.10451 | | | gene-MA16_Dca017504;gene-MA16_Dca026687 |
| | GO:0005797 | Golgi medial cisterna | 0.16459 | | | MA16_Dca001958 |
| | GO:0005852 | eukaryotic translation initiation factor 3 complex | 0.16459 | | | gene-MA16_Dca001383;gene-MA16_Dca001386 |
| | GO:0005871 | kinesin complex | 0.16459 | | | gene-MA16_Dca013578;gene-MA16_Dca026695 |
| | GO:0016020 | membrane | | 0.00913 | | gene-MA16_Dca000800;gene-MA16_Dca000921;gene-MA16_Dca002727; gene-MA16_Dca002728;gene-MA16_Dca003087;gene-MA16_Dca005684; gene-MA16_Dca006160;gene-MA16_Dca007647;gene-MA16_Dca008165; gene-MA16_Dca009282;gene-MA16_Dca009818;gene-MA16_Dca012412; gene-MA16_Dca013211;gene-MA16_Dca013212;gene-MA16_Dca013412; gene-MA16_Dca014148;gene-MA16_Dca014149;gene-MA16_Dca015954; gene-MA16_Dca016040;gene-MA16_Dca018716;gene-MA16_Dca019398; gene-MA16_Dca019399;gene-MA16_Dca019400;gene-MA16_Dca019934; gene-MA16_Dca019958;gene-MA16_Dca021265;gene-MA16_Dca021557; gene-MA16_Dca021561;gene-MA16_Dca022112;gene-MA16_Dca024381; gene-MA16_Dca025322 |

Li et al. (2021), *PeerJ*, DOI 10.7717/peerj.10769

**Table 3** (*continued*)

| ID | Description | q value | | | geneID |
|---|---|---|---|---|---|
| | | **FR1-CK** | **FR4-CK** | **FR4-FR1** | |
| GO:0000110 | nucleotide-excision repair factor 1 complex | | 0.19328 | | gene-MA16_Dca007683 |
| GO:0000153 | cytoplasmic ubiquitin ligase complex | | 0.19328 | | gene-MA16_Dca006393 |
| GO:0005672 | transcription factor TFIIA complex | | 0.19328 | 0.17955 | gene-MA16_Dca004588 |
| GO:0009706 | chloroplast inner membrane | | 0.24481 | | gene-MA16_Dca015955;gene-MA16_Dca016038 |
| GO:0005801 | cis-Golgi network | | | 0.09257 | gene-MA16_Dca000620;gene-MA16_Dca001958 |
| GO:0005783 | endoplasmic reticulum | | | 0.17955 | gene-MA16_Dca003087;gene-MA16_Dca015315;gene-MA16_Dca015955; gene-MA16_Dca017315;gene-MA16_Dca018760;gene-MA16_Dca018761; gene-MA16_Dca024381 |
| GO:0005797 | Golgi medial cisterna | | | 0.17955 | gene-MA16_Dca001958 |
| GO:0005622 | intracellular | | | 0.17955 | gene-MA16_Dca012408;gene-MA16_Dca012411;gene-MA16_Dca013217; gene-MA16_Dca015311;gene-MA16_Dca015854;gene-MA16_Dca015855; gene-MA16_Dca018978;gene-MA16_Dca022871 |

Li et al. (2021), *PeerJ*, DOI 10.7717/peerj.10769

**Table 3** (*continued*)

| | ID | Description | q value | | | geneID |
|---|---|---|---|---|---|---|
| | | | **FR1-CK** | **FR4-CK** | **FR4-FR1** | |
| **Molecular function** | GO:0003743 | translation initiation factor activity | 0.00214 | | | gene-MA16_Dca001383;gene-MA16_Dca001386;gene-MA16_Dca002840; gene-MA16_Dca018345;gene-MA16_Dca018926;gene-MA16_Dca018927; gene-MA16_Dca026689 |
| | GO:0010012 | steroid 22-alpha hydroxylase activity | 0.00543 | | | gene-MA16_Dca015315;gene-MA16_Dca018761 |
| | GO:0045735 | nutrient reservoir activity | 0.00543 | | | gene-MA16_Dca023265;gene-MA16_Dca023266;gene-MA16_Dca023267 |
| | GO:0030145 | manganese ion binding | 0.02937 | | | gene-MA16_Dca023265;gene-MA16_Dca023266;gene-MA16_Dca023267 |
| | GO:0003954 | NADH dehydrogenase activity | 0.02937 | 0.06782 | | gene-MA16_Dca014146;gene-MA16_Dca014147 |
| | GO:0008171 | O-methyltransferase activity | | 0.01400 | | gene-MA16_Dca007905;gene-MA16_Dca007906;gene-MA16_Dca007907; gene-MA16_Dca007908 |
| | GO:0004848 | ureidoglycolate hydrolase activity | | 0.01400 | | gene-MA16_Dca005252;gene-MA16_Dca005253 |
| | GO:0008373 | sialyltransferase activity | | 0.05484 | | gene-MA16_Dca009824;gene-MA16_Dca013452 |
| | GO:0005388 | calcium-transporting ATPase activity | | 0.12133 | 0.02220 | gene-MA16_Dca015955;gene-MA16_Dca021262 |
| | GO:0015267 | channel activity | | | 0.00196 | gene-MA16_Dca017309;gene-MA16_Dca017310;gene-MA16_Dca017311 |
| | GO:0010012 | steroid 22-alpha hydroxylase activity | | | 0.01496 | gene-MA16_Dca015315;gene-MA16_Dca018761 |
| | GO:0003865 | 3-oxo-5-alpha-steroid 4-dehydrogenase activity | | | 0.02220 | gene-MA16_Dca007792;gene-MA16_Dca007793 |
| | GO:0019166 | trans-2-enoyl-CoA reductase (NADPH) activity | | | 0.03513 | gene-MA16_Dca007792;gene-MA16_Dca007793 |

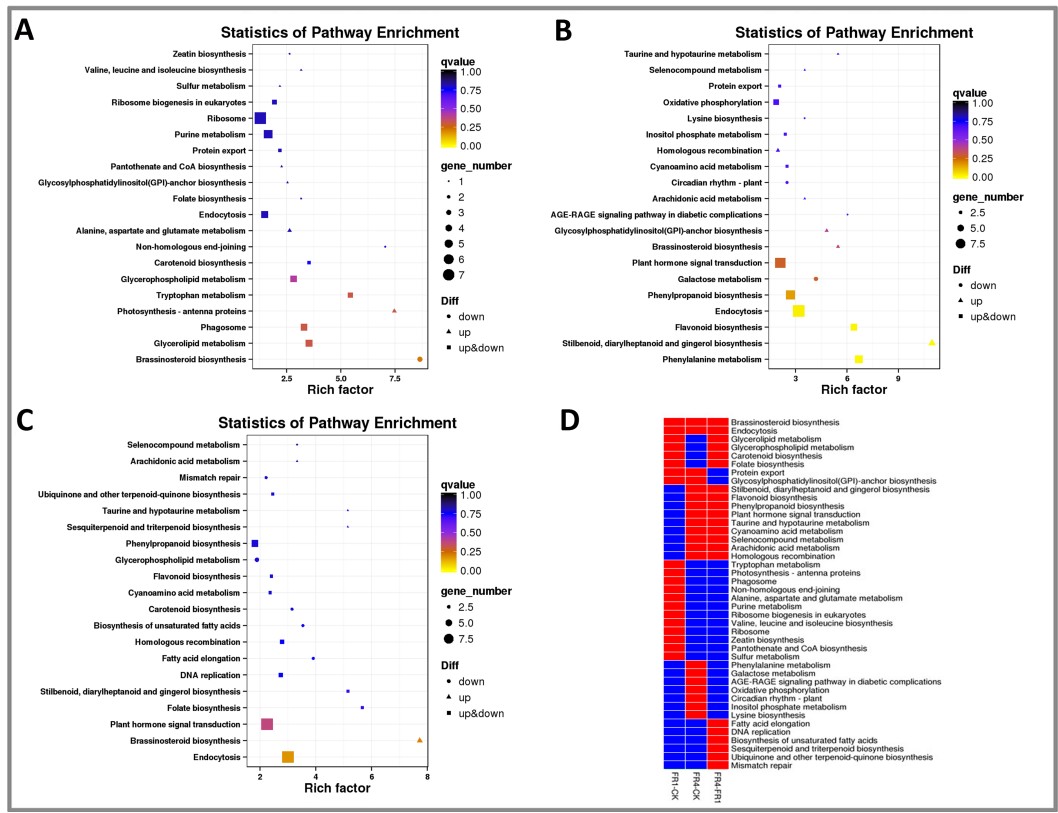

**Figure 3** **KEGG enrichment analysis of the target genes of DE lncRNAs in *D. officinale* under different light treatments.** (A) FR1-CK. (B) FR4-CK. (C) FR4-FR1. (D) Top 20 KEGG pathways enriched in target genes of DE lncRNAs in the three groups. The red colour indicates that the comparison contains the indicated pathway, and the blue colour indicates that the comparison does not contain the pathway.

## KEGG enrichment analysis of DE lncRNAs in *D. officinale*

A KEGG enrichment analysis of the target genes of the DE lncRNAs was also performed (Fig. 3). The most enriched pathways obtained from the analysis of the FR1-CK comparison included brassinosteroid biosynthesis, photosynthesis-antenna proteins, nonhomologous end-joining, tryptophan metabolism and glycerolipid metabolism (Fig. 3A). In contrast, the top five enriched pathways identified from the results of the FR4-CK comparison included stilbenoid, diarylheptanoid and gingerol biosynthesis, phenylalanine metabolism, flavonoid biosynthesis, the AGE-RAGE signalling pathway in diabetic complications, and taurine and hypotaurine metabolism (Fig. 3B). The top five enriched pathways obtained from the FR4-FR1 comparison included brassinosteroid biosynthesis, folate biosynthesis, stilbenoid, diarylheptanoid and gingerol biosynthesis, taurine and hypotaurine metabolism, and sesquiterpenoid and triterpenoid biosynthesis (Fig. 3C).

The top 20 enriched pathways identified from the results of the three comparisons (FR1-CK, FR4-CK, and FR4-FR1) were also analysed, and some of the above-mentioned enriched pathways, such as brassinosteroid biosynthesis and endocytosis, were found in these set of top 20 pathways three comparisons (Fig. 3D). These results indicated that these

pathways exhibit significant differences under the three different treatments (CK, FR1 and FR4).

Moreover, several of the top 20 enriched pathways were also detected in two of the three comparisons (FR1-CK and FR4-FR1), and these included glycerolipid metabolism, glycerophospholipid metabolism, carotenoid biosynthesis and folate biosynthesis (Fig. 3D), which indicates that the FR1 treatment might have a significant effect on these pathways. Protein export and glycosylphosphatidylinositol (GPI)-anchor biosynthesis were also identified among the top 20 pathways obtained from the FR1-CK and FR4-CK comparisons (Fig. 3D), which indicated that protein export and GPI-anchor biosynthesis might play important roles in the response to far-red light. The following pathways were among the top 20 enriched pathways identified from the FR4-CK and FR4-FR1 comparisons: stilbenoid, diarylheptanoid and gingerol biosynthesis, flavonoid biosynthesis, phenylpropanoid biosynthesis, plant hormone signal transduction, and taurine and hypotaurine metabolism (Fig. 3D). These results indicate that the FR4 treatment might exert significant effects on these pathways.

Moreover, some of the top 20 enriched pathways were among the top 20 pathways identified from the results of a single comparison; for example, the tryptophan and sulphur metabolism pathways were among the top 20 pathways obtained for the FR1-CK comparison but were not found in the top 20 pathways found for the other two comparisons (Fig. 3D), which indicated that the FR1 treatment might significantly affect tryptophan metabolism and sulphur metabolism. The phenylalanine metabolism, lysine biosynthesis, and circadian rhythm-plant pathways were among the top 20 enriched pathways found only for the FR4-CK comparison (Fig. 3D), which indicated that the FR4 treatment exerts a significant effect on these pathways. Similarly, sesquiterpenoid and triterpenoid biosynthesis were among the top 20 pathways identified from the FR4-FR1 comparison, which indicated that the FR4 treatment has a stronger effect on sesquiterpenoid and triterpenoid biosynthesis than does the FR1 treatment.

## Network of far-red light-responsive lncRNAs and their targets

To further understand the function of DE lncRNAs in the response of *D. officinale* to far-red light, a lncRNA-mRNA interaction network was constructed using Cytoscape software (v.3.6.0) (Figs. 4A-4B; Tables S12–S13).

Among the lncRNAs obtained from the FR1-CK comparison, MSTRG.60454.1 had the highest number of target genes (18), and these are mainly involved in ribosome biogenesis in eukaryotes, sulphate assimilation, the cytoplasm, membranes, and protein processing in the endoplasmic reticulum (Fig. 4A; Table S12). The expression of MSTRG.41036.9, which is mainly involved in sulphur metabolism, the mitochondrial inner membrane, phosphorylation and RNA transport, exhibited the most significant upregulation (Fig. 4A; Table S12), whereas that expression of MSTRG.70733.3, which is mainly involved in hydrolase activity and activity on ester bonds, showed the most significant downregulation (Fig. 4A; Table S12). In addition, some target genes were also found to be regulated by different lncRNAs identified from the FR1-CK comparison. For example, the genes MA16_Dca022892, MA16_Dca022894, and MA16_Dca022895 are jointly regulated by

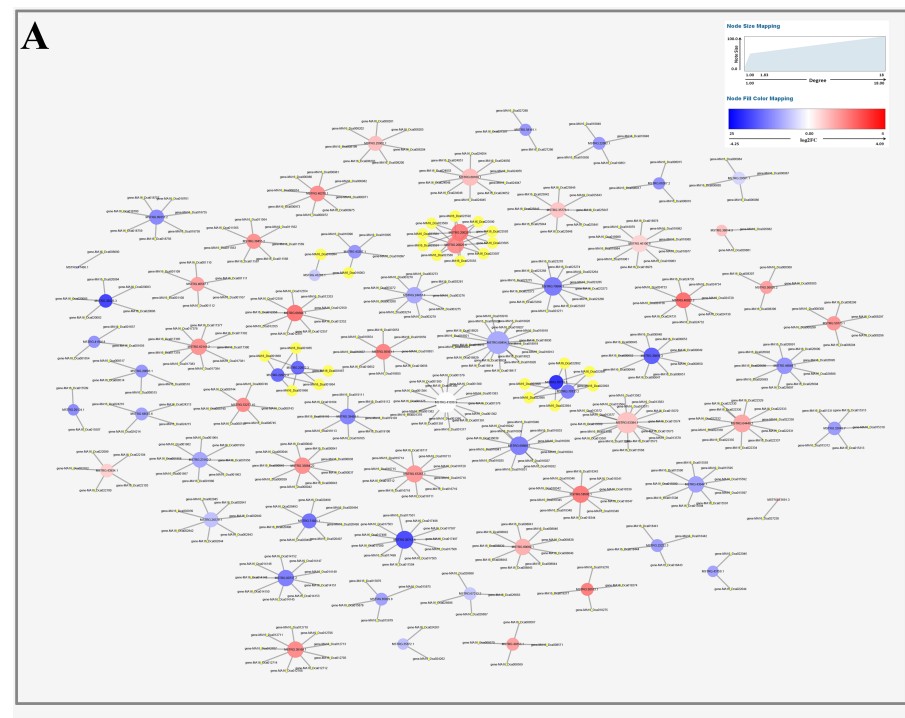

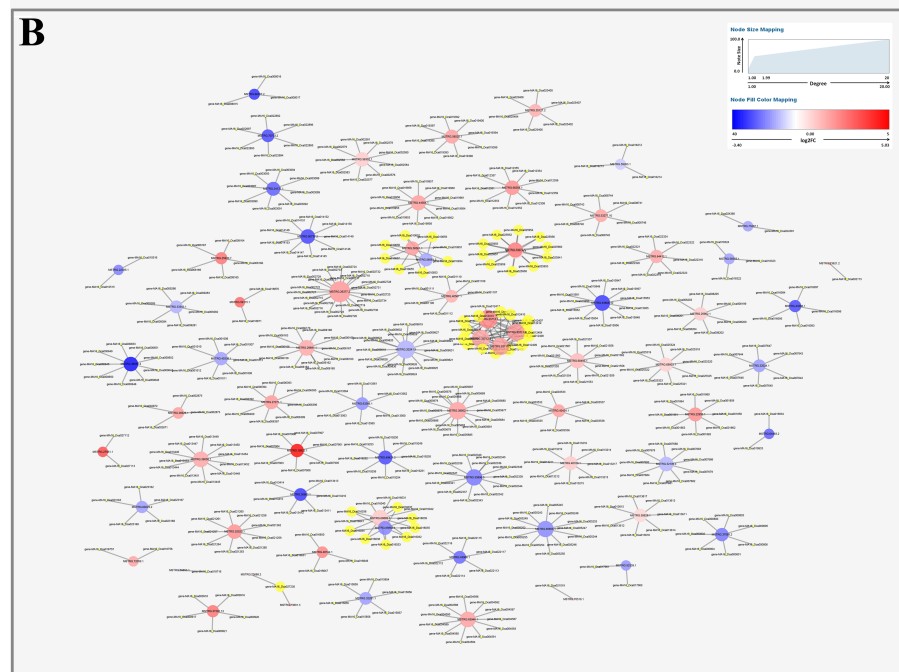

**Figure 4 The relationship prediction of DE lncRNAs and their targets in *D. officinale* under different light treatments.** (A) Interaction network of the results from the FR1-CK comparison. (B) Interaction network of the results from the FR4-CK. The gradient-coloured circle nodes represent lncRNAs, and the blue circular nodes represent mRNAs. High expression and low expression of lncRNAs is shown in dark red and dark green respectively. The size of the circle indicates the number of connected genes. The solid lines indicate interaction associations between lncRNAs and mRNAs.

MSTRG.70733.3 and MSTRG.70733.2 (Fig. 4A). Therefore, the above-mentioned lncRNAs are most likely involved in the effects of FR1 on *D. officinale*.

Among the lncRNAs obtained from the FR4-CK comparison, MSTRG.26377.2 had the maximal number of target genes (20), and these genes are mainly involved in plant hormone signal transduction, chloroplasts, membranes, and pyruvate metabolism (Fig. 4B; Table S13). MSTRG.19522.1, which is mainly involved in O-methyltransferase activity, methylation, phenylpropanoid biosynthesis, flavonoid biosynthesis, and stilbenoid, diarylheptanoid and gingerol biosynthesis, exhibited the most significantly upregulated expression (Fig. 4B; Table S13), and the lncRNA that showed the most significant downregulation was MSTRG.36859.3, which is mainly involved in the plant circadian rhythm (Fig. 4B; Table S13). In addition, genes such as MA16_Dca1241, MA16_Dca12410, and MA16_Dca12404 were regulated by MSTRG.33712.1, MSTRG.33712.2, MSTRG.33712.3 and MSTRG.33712.5 (Fig. 4B). Therefore, the above-mentioned lncRNAs are most likely involved in the effects of the FR4 treatment on *D. officinale*.

## Secondary metabolite contents in *D. officinale* under different light treatments

The transcriptome sequencing revealed that flavonoid metabolism, alkaloid metabolism, and carotenoid metabolism might affect the shade-avoidance response of *D. officinale* under far-red light. Therefore, the flavonoid, alkaloid and carotenoid contents in the leaves and stems of *D. officinale* were measured after the different far-red light treatments (Fig. 5). The highest flavonoid content in *D. officinale* leaves was obtained after the FR4 treatment (38.121 mg g$^{-1}$), followed by the FR1 treatment (32.343 mg g$^{-1}$), whereas the CK treatment yielded the lowest flavonoid content (29.860 mg g$^{-1}$) (Fig. 5A; Table S14). The highest flavonoid content in the stems was detected after the FR4 treatment (20.249 mg g$^{-1}$), followed by the FR1 treatment (18.641 mg g$^{-1}$), whereas the lowest content was observed after the CK treatment (13.875 mg g$^{-1}$) (Fig. 5B; Table S15). The highest leaf alkaloid content was obtained with the FR4 treatment, followed by the FR1 treatment, whereas the lowest content was found with the CK treatment (Fig. 5C; Table S16). The FR4 and FR1 treatments yielded the highest alkaloid content in the stem segments, whereas the lowest content was detected with the CK treatment (Fig. 5D; Table S17). The highest carotenoid content in the leaves and stems was obtained with the FR4 treatment, followed by the FR1 treatment, whereas the lowest content was found with the CK treatment (Figs. 5E– 5F; Tables S18–S19). The highest polysaccharide content of leaves was obtained after the FR4 treatment (108.995 mg g$^{-1}$), followed by the FR1 treatment (101.683 mg g$^{-1}$), whereas the CK treatment yielded the lowest polysaccharide content (94.086 mg g$^{-1}$) (Fig. 5G; Table S20). The highest polysaccharide content of stems was detected after the FR1 treatment (83.294 mg g$^{-1}$), followed by the FR4 treatment (80.241 mg g$^{-1}$), whereas the lowest content was observed after the CK treatment (68.811 mg g$^{-1}$)

(Fig. 5H; Table S21). Therefore, far-red light at appropriate proportions can promote the accumulation of flavonoids, alkaloids, carotenoids and polysaccharides in *D. officinale*.

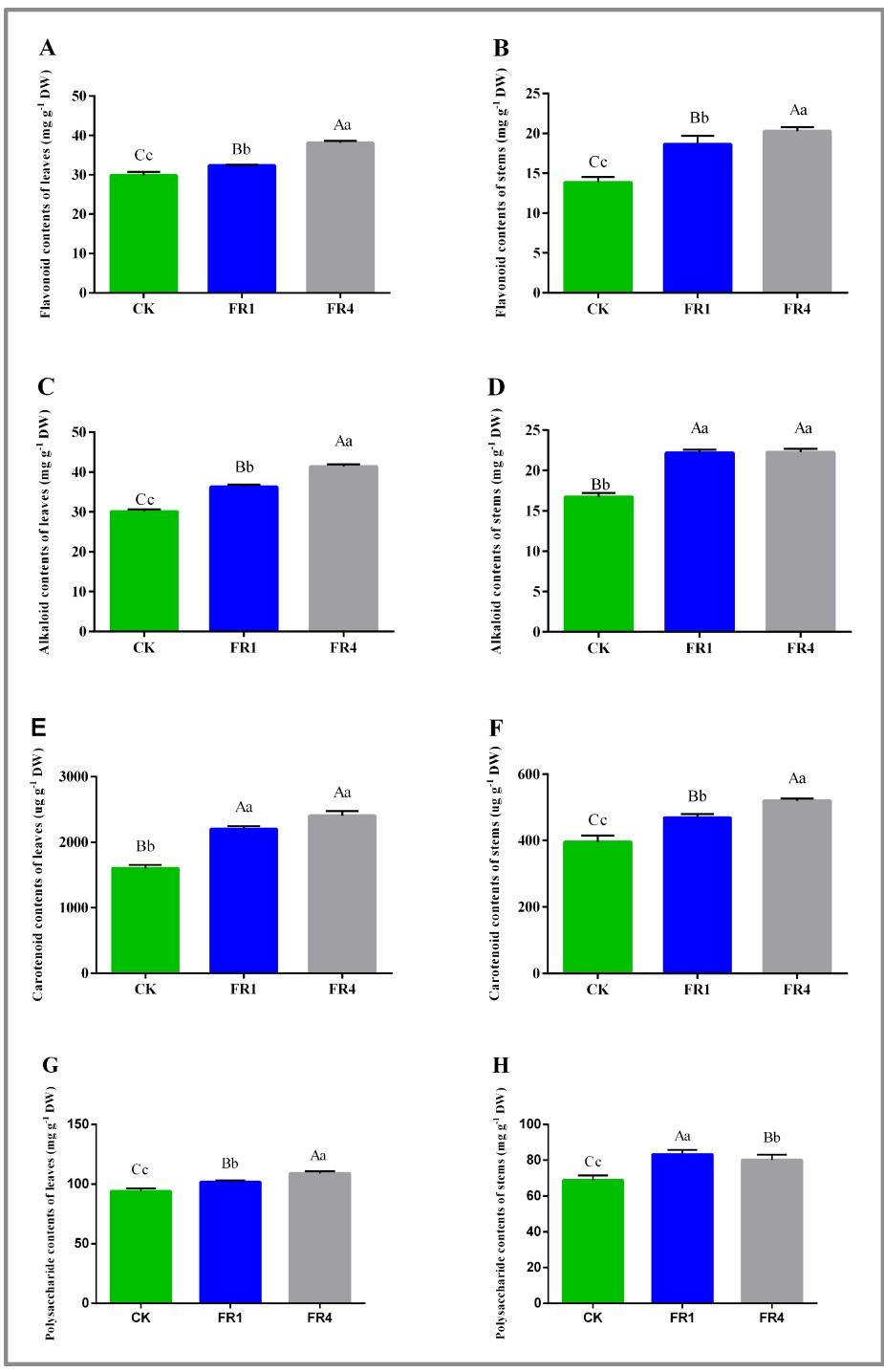

**Figure 5  Metabolite contents in *D. officinale* under different light treatments.** (A) and (B) represent changes in the flavonoid contents in the leaves and stems respectively. (C) and (D) represent changes in the alkaloid contents in the leaves and stems respectively. (E) and (F) represent changes in the carotenoid contents in the leaves and stems respectively. (G) and (H) represent changes in the polysaccharide contents in the leaves and stems respectively. Different upper/lowercase letters indicate statistically significant differences at the 0.01/0.05 level, as determined by one-way ANOVA and Duncan's test.

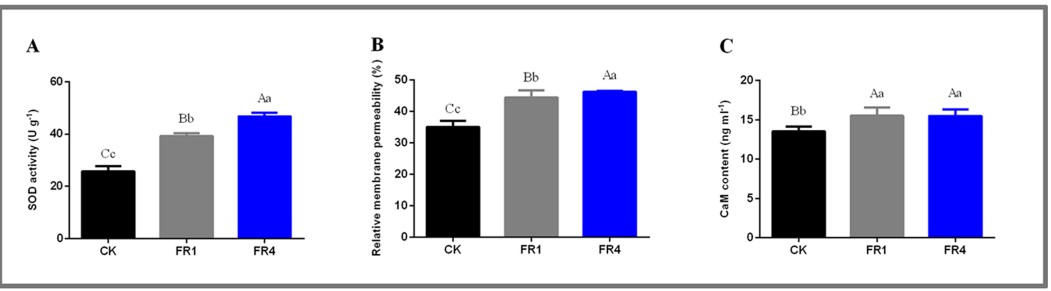

**Figure 6** **Levels of physiological and biochemical indicators in the leaves of *D. officinale* under different light treatments.** (A) SOD activity; (B) Relative membrane permeability; (C) CaM content.

## Levels of physiological and biochemical indicators in *D. officinale* under different light treatments

The transcriptome sequencing revealed that membrane and calcium-transporting ATPase activity might affect the shade-avoidance response of *D. officinale* under far-red light. Therefore, the levels of related physiological and biochemical indicators in the leaves of *D. officinale* under the different far-red light treatments were measured. As an important active oxygen scavenger, SOD can participate in the cell membrane lipid peroxidation defence system. The highest SOD activity was detected with the FR4 treatment, followed by the FR1 and CK treatments (Fig. 6A; Table S22). The highest cell membrane permeability was observed after the FR4 treatment, followed by the FR1 and CK treatments (Fig. 6B; Table S23). The CaM contents obtained with the FR4 and FR1 treatments were higher than that detected after the CK treatment (Fig. 6C; Table S24). These results indicated that the effect of far-red light on *D. officinale* might be closely related to the cell membrane and $Ca^{2+}$ transduction.

## Identification of DE lncRNAs and their targets in *D. officinale* under different light treatments by qPCR

The qPCR analysis performed in this study revealed 12 groups of lncRNAs and target genes (Fig. 7; Tables S25–S27). Only the expression pattern of MSTRG.58590.1 and its target gene *TTA*1, MSTRG.19522.1 and its target gene MA16_Dca007905, and MSTRG.38867.1 and its target gene *PHYA*1 were negatively correlated, which indicated that these three target genes were negatively regulated by their corresponding lncRNAs. MSTRG.63384.1 and its target gene *ZSD*1, MSTRG.29691.1 and its target gene *GGT*1_5, MSTRG.66273.1 and its target gene *COP*1, and MSTRG.48624.1 and its target gene *HY*5 were positively correlated, which indicating that these four target genes were positively regulated by their corresponding lncRNAs. The remaining lncRNAs and target genes showed different regulatory patterns, which might explain why the target genes were regulated by different members of lncRNA families or other lncRNAs in the plant responses to different light conditions.

Some DE lncRNA target genes are related to metabolic pathways. For example, the target gene *TAA*1 of MSTRG.58590.1 is a key gene of the tryptophan metabolism pathway, and the expression level of *TAA*1 after the FR4 treatment was higher than the levels obtained after

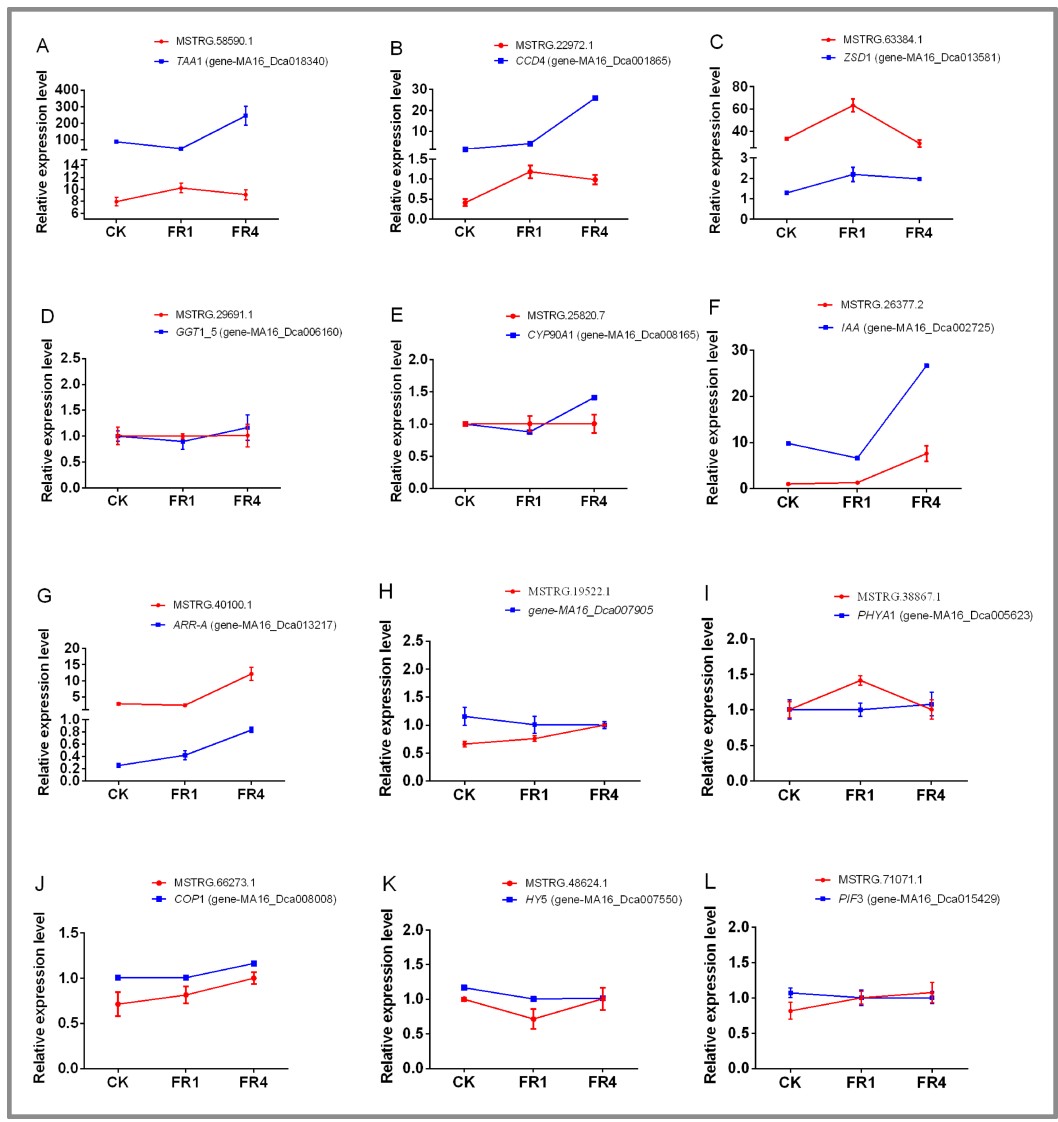

**Figure 7** **Identification of DE lncRNAs and their target genes in *D. officinale* under different far-red light conditions via qPCR.** *TAA*1, tryptophan aminotransferase 1; *CCD*4, carotenoid cleavage dioxygenase; *ZSD*1, zerumbone synthase; *GGT*1_5, gamma-glutamyl transpeptidase 3-like; *CYP*90*A*1, cytochrome P450 family 90 subfamily A polypeptide 1; *IAA*, auxin-responsive protein IAA; *ARR-A*, two-component response regulator ARR-A family; *PHY*A, phytochrome A; *COP*1, constitutively photomorphogenic 1; *HY*5, elongated hypocotyl 5; *PIF*3, phytochrome-interacting factor 3.

the FR1 and CK treatment (Fig. 7A). The target gene *CCD*4 of MSTRG.22972.1 and the target gene *ZSD*1 of MSTRG.63384.1 are key genes of the carotenoid synthesis pathway. The expression levels of *CCD*4 and *ZSD*1 under far-red light were higher than those obtained after the CK treatment (Figs. 7B-7C). Some DE lncRNA target genes are related to hormone signal transduction. The target gene *CYP* 90*A*1 of MSTRG.25820.7 is involved in the brassinolide signal transduction pathway in *D. officinale*. The highest expression of *CYP*90*A*1 was obtained with the FR4 treatment, followed by the FR1 and CK treatments,

which indicated that brassinolide signal transduction plays an important role in the shade-avoidance response of *D. officinale* under far-red light (Fig. 7E). Some DE lncRNA target genes are closely related to DNA methylation. For example, the MSTRG.19522.1 target gene MA16_Dca007905 functions in O-methyltransferase activity (Fig. 7H). In addition, some DE lncRNA target genes are also related to far-red light signal transduction; for example, MSTRG.38867.1 and its target gene *PHYA*1, MSTRG.69319.1 and its target gene *SPA*1, MSTRG.66273.1 and its target gene *COP*1, MSTRG.48624.1 and its target gene *HY* 5 MSTRG.71071.1 and its target gene *PIF*3 might be involved in the signal transduction pathways of *D. officinale* induced in response to far-red light (Figs. 7I-7L).

## DISCUSSION

### Cell signalling perception and conduction in *D. officinale* under far-red light

The GO analysis revealed that the lncRNAs identified from the FR4-CK comparison were significantly enriched in the membrane, and the lncRNAs identified from the FR1-CK and FR4-CK comparisons were significantly enriched in calcium-transporting ATPase activity (Table 3). The physiological and biochemical experiments also revealed that the activity of SOD, the cell membrane permeability and the CaM content obtained after far-red light treatment were higher than those obtained after the control treatment (Figs. 6A–6C). Therefore, the shade-avoidance response of *D. officinale* to far-red light might be closely related to the cell membrane and $Ca^{2+}$ transduction. The exposure of plant cells to external factors induce corresponding changes in the cell wall components. Elements that perceive environmental changes are distributed in the periplasmic interval between the cell wall and the cell membrane, and these elements can induce increases in the $Ca^{2+}$ concentration in the cytosol. The cell membrane carries information concerning external factors as well as cell information and participates in energy and material exchange. Moreover, the cell membrane can maintain the stability of the intracellular environment and provide order to physiological and metabolic pathways in plants (*Graier et al., 2010*). As an important secondary messenger in plants, $Ca^{2+}$ can respond to external factors (*Moeder, Phan & Yoshioka, 2019*). Under the influence of external factors, the $Ca^{2+}$ concentration of plants increases instantaneously and can be sensed by various $Ca^{2+}$ sensors and binding proteins, including calmodulin (CaM), calcium-dependent protein kinases (CDPKs) and calcineurin B-like proteins (CBLs) (*González, Sáez & Moenne, 2018*). CaM can act on various organelles and their constituents, such as the cell membrane, vacuole, and nucleus, and is involved in the regulation of important physiological and biochemical processes and secondary metabolism in plants (*Yamashita et al., 2018*). For example, CaM plays an important role in the reproductive growth of plants, and its gene family members CaM2 and CaM7 can promote the growth of pollen tubes. *Wang et al. (2010a)* and *Wang et al. (2010b)* reported that CaM is involved in potato glycoside alkaloid metabolism in response to light.

## lncRNAs involved in some metabolic pathways influence the effects of far-red light on the shade-avoidance response of *D. officinale*

The KEGG analysis revealed that the following pathways were among the top 20 pathways enriched in the lncRNAs identified from the FR1-CK and FR4-CK comparisons: phenylalanine metabolism, phenylpropanoid biosynthesis, tryptophan metabolism, stilbenoid, diarylheptanoid and gingerol biosynthesis, flavonoid biosynthesis, and carotenoid biosynthesis (Fig. 3). The phenylalanine metabolism and phenylpropanoid biosynthesis pathways are upstream of tryptophan metabolism, stilbenoid, diarylheptanoid and gingerol biosynthesis, flavonoid biosynthesis, and carotenoid biosynthesis (*Li et al., 2019*). Physiological and biochemical experiments also verified that the contents of flavonoids, carotenoids, alkaloids, and polysaccharides after the far-red light treatments were higher than those obtained after the control treatment.

The results of this study showed that far-red light can promote the accumulation of flavonoids in *D. officinale* because phytochrome A (*PHYA*) is an essential component in far-red light-induced flavonoid synthesis (*Li et al., 2014*). Plant flavonoid synthesis is related to the response to external factors, including far-red light (*Li et al., 2014*), and flavonoids protect plants from oxidative damage by absorbing far-red light and removing reactive oxygen species (*Xie et al., 2015*). Far-red light induces plant photomorphogenesis and flavonoid accumulation through the inhibition of CUL4-DDB1$^{COP1/SPA}$ by *PHYA* to stabilize the degradation of its substrate (*Maier & Hoecker, 2015*).

The results of this study also showed that tryptophan metabolism (upstream of alkaloid metabolism) was significantly enriched (Fig. 3), and the expression of *TAA*1 (DE lncRNA target gene), which is involved in the synthesis of metabolites, was upregulated in *D. officinale* under far-red light treatment (Fig. 7). Studies on *Arabidopsis* have revealed that *TAA*1 is involved in catalysing the reaction of tryptophan and pyruvate to produce etodolac and, subsequently, auxin (*Won et al., 2011*). The discovery of *TAA*1 confirmed the existence of a pathway for the synthesis of indoleacetic acid (IAA) in plants. In addition, after perceiving shaded signals, *Arabidopsis* plants might rapidly increase their level of IAA through the regulation of *TAA*1 and thereby rapidly induce the expression of IAA response genes and downstream genes, which leads to rapid hypocotyl extension, among other effects that form part of the shade-avoidance phenomena (*Won et al., 2011*). Therefore, tryptophan metabolism might affect the shade-avoidance response of *D. officinale* under far-red light.

The results of this study also showed that carotenoid metabolism was significantly enriched in *D. officinale* (Fig. 3). The expression of both the *CCD*4 and *ZSD*1 genes (DE lncRNA targets) was upregulated under far-red light treatment (Fig. 7). *CCD*4 is a key gene in apocarotenoid synthesis, and *ZSD*1 is a key gene in ABA synthesis. Related studies have shown that strawberry leaves have a high cucurbitacin content under low R:FR ratio (*Sun, Yue-Fei & Lu-Yi, 2010*). Carotenoids are an indispensable class of pigments in higher plants. These compounds act as antenna pigments for photosynthesis and transmit captured light energy to chlorophyll. Carotenoids also function in light protection and the scavenging of free radicals and thus protect plant tissues from damage induced by strong light (*Sun, Yue-Fei & Lu-Yi, 2010*). Carotenoids are also synthetic precursors of

phytohormones such as ABA, strigolactone and lactone, which play important roles in plant growth and development (*Liu et al., 2016*). Modern medical research has shown that carotenoids are strong antioxidants that can protect cells from oxidative damage and maintain the normal function of the cellular system (*Liu et al., 2016*). Therefore, carotenoid metabolism might affect the shade-avoidance response of *D. officinale* under far-red light.

The KEGG analysis revealed the AGE-RAGE signaling pathway in diabetic complications as among the top 5 pathways enriched in the lncRNAs identified from the FR4-CK comparisons (Fig. 3D). The AGE-RAGE signaling pathway plays an important role in the occurrence and development of diabetic nephropathy (*Yang, Yang & Ma, 2019*). Modern pharmacological studies have shown that Coptis polysaccharide can significantly reduce the expression of HU-VEC on mRNA and protein RAGE induced by AGEs (*Yin et al., 2012*). It is confirmed that Coptis polysaccharide might treat diabetic nephropathy by blocking the AGE-RAGE signaling pathway (*Yin et al., 2012*). *Yuan et al. (2017)* confirmed that puerarin has a significant inhibitory effect on the formation of AGEs in vivo and in vitro, and reduce the expression of RAGE on mRNA in the kidney tissue of streptozotocin-induced diabetic rats to improve diabetic nephropathy (*Shen et al., 2009*). Related studies also show that berberine can significantly affect the AGE-RAGE signaling pathway, and its important target factor is AGEs (*Qiu, Tang & Wei, 2017*). The AGE- RAGE signaling pathway is significantly enriched in the shade-avoidance response of *D.officinale* under far-red light, which may be closely related to the metabolic pathways of polysaccharides, flavonoids and alkaloids.

In addition, the KEGG analysis revealed taurine and hypotaurine metabolism as among the top 20 pathways enriched in the lncRNAs identified from the FR4-CK and FR4-FR1 comparisons (Fig. 3D). Hypotaurine is an $S^{2-}$ scavenger, which plays an important role in sulfur metabolism in plants (*Shan et al., 2011*). $H_2S$ is produced by the decomposition of cysteine in plants, and cysteine is an important hub of the sulfur conversion pathway. Cysteine can also be broken down to produce pyruvic acid, which is an important substance in carbon metabolism (*Shan et al., 2011*). Studies on corn have found that $H_2S$ can promote the synthesis of triterpenoids by regulating sulfur and carbon metabolism(*Capaldi et al., 2015*). During the response of *D. officinale* to far-red light, the expression of $\gamma$-glutamyl transpeptidase (*GGT*1, DE lncRNA target gene), which is involved in the taurine and sulfinate metabolic pathways, was significantly upregulated. Increase in the content of 5-glutamyl taurine increased were accompanied by decreases in the taurine and sulfinate (a $S^{2-}$ scavenger) contents and increase in the content of $S^{2-}$. Increased $S^{2-}$ concentrations might affect the mevalonate pathway as well as the synthesis of alkaloids, carotenoids, and fatty acids, among other metabolites (*Mugford et al., 2011*). Therefore, taurine and hypotaurine metabolism might play an important role in the shade-avoidance response of *D. officinale* under far-red light.

## lncRNAs involved in hormone signal transduction influence the shade-avoidance response of *D. officinale* to far-red light

During the response of *D. officinale* to far-red light, the functions of target genes of DE lncRNAs involve plant hormone signal transduction (Fig. 3; Table 3). Light signals are

closely related to hormones during the process of regulating plant growth and development (*Warpeha & Montgomery, 2015*). The plant hormone levels exhibit changes depending on the light conditions and photosensitive pigment activity, which indicates that light signals can participate in plant hormone signal transduction and thereby regulate plant growth and development and the activity of metabolic pathways (*Li et al., 2018*). The plant shade-avoidance response also involves hormone regulation, and the GO and KEGG enrichment analyses performed in this study showed that BR, auxin and cytokinin were significantly enriched in *D. officinale* under far-red light (Fig. 3; Table 3).

BR can promote plant cell elongation, division and vascular bundle development. BR plays an important role in regulating the light response of plants (*Nafie, 2015*). BR can also activate the antioxidant enzyme protection system and thereby eliminates excessive harmful free radicals induced as a result of external factors and improves the ability of plants to respond to external factors (*Oh, Zhu & Wang, 2012*). The results of this study showed that the expression of BR signalling kinase (*BSK*, DE lncRNA target gene), a gene downstream of BRI1, was significantly upregulated by the FR4 treatment (Fig. 7). Related studies have shown that brassinosteroid insensitive 1 (*BRI*1) is a BR receptor with kinase activity. The cultivation of *Arabidopsis* plants under far-red light rapidly upregulates the expression of *BRI*1. *Arabidopsis* plants without *BRI*1 function comprise significantly fewer cells and are severely dwarfed. These findings show that *BRI*1 plays an important role in the light-induced elongation of *Arabidopsis* (*Ouyang et al., 2011*).

Auxin (IAA) is a hormone that regulates the direction of plant growth resulting from specific regions of cells presenting increased activity and division.The main role of IAA is to regulate plant growth, and this hormone specifically stimulates the vertical growth of stem cells and inhibits the lateral growth of root cells (*Gan et al., 2013*). Studies in *Arabidopsis* have shown that the *TAA*1, indole-3-acetic acid inducible (*IAA*), and pin-formed (*PIN*) genes participate in the shade-avoidance response (*Yi et al., 2008*). Moreover, the *TAA*1 mutant sav3 exhibits the hypoxia hypothesis because the hypocotyl elongated faster than that of the wild type under a low R:FR ratio. *IAA* is an early auxin response gene and encodes a transcription factor that regulates auxin expression. Researchers have found that the expression of *IAA*19 under low R:FR conditions is three-fold higher than that under white light (*Pierik et al., 2009*). The results of this study showed that the expression levels of *TAA*1 and *IAA* (DE lncRNA targets) in the auxin signalling pathway were significantly upregulated under far-red light (Fig. 7); thus, these genes might be key players in the effects of far-red light on the shade-avoidance response of *D. officinale*.

Cytokinins can promote cell division and differentiation, eliminate apical dominance, and promote lateral bud growth. Related reports indicate that cytokinins are involved in the shade-avoidance response of *A. thaliana*, and the genes involved in the shade-avoidance response include cytokinin oxidase 5 (*CKX*5) and cytokinin oxidase 6 (*CKX*6), both of whose encoded proteins can degrade key enzymes involved in cytokinin synthesis and regulate the cytokinin levels. Researchers using the $\beta$-glucuronidase (GUS) staining method found that the expression of AtCKX6::GUS is significantly increased in *Arabidopsis* protolayer cells of the leaf primordium under low R:FR conditions (*Carabelli et al., 2007*).

The results of this study showed that the expression of *ARR-A* (which encodes a two-component response regulator of the ARR-A family, DE lncRNA target gene) in the cytokinin signalling pathway was significantly upregulated under far-red light (Fig. 7), which might affect the shade-avoidance response of *D. officinale* under far-red light.

In summary, lncRNAs might be involved in the signal transduction of BR, auxin and cytokinins and thereby influence the effects of far-red light on the shade-avoidance response of *D. officinale*.

## DNA methylation affects the shade-avoidance response of *D. officinale* under far-red light

The functions of some target genes of DE lncRNAs involve O-methyltransferase activity, and the GO analysis revealed that the lncRNAs identified in the FR4-CK comparison were significantly enriched in this activity (Table 3). DNA methylation is an important epigenetic modification that plays an important role in shaping the chromatin structure and regulating gene expression. DNA methylation is involved in many biological activities, such as embryonic development, abiotic stress, light regulation, and tissue-specific gene expression (*Vanyushin & Ashapkin, 2011*; *Kuo et al., 2015*). *Kuo et al. (2015)* used whole-genome expression and DNA methylation profiles to study the effects of red and far-red light on genes regulated by the secondary metabolic pathways in agarwood and found that the expression of small RNA-regulated genes might occur through RNA-directed DNA methylation. Laser radiation at a certain frequency not only causes changes in the DNA methylation patterns in rice (*O. sativa* L.) but also alters the expression of epigenetically modified genes that function in the maintenance, structure and state of chromatin, such as causing changes in the expression of methyltransferases, glycosidases and important enzymes involved in the short interfering RNA (siRNA) production pathway. Moreover, a correlation analysis confirmed that these changes are related to methylation *(Li et al., 2017)*. *Wang et al. (2010a)*; *Wang et al. (2010b)* treated geminated seeds of orthogonal sorghum F1 hybrids and pure parents with a low-intensity laser and found that the sorghum F1 hybrids and corresponding parents presented some changes in the level and pattern of DNA methylation. Therefore, O-methyltransferase activity might play an important role in the shade-avoidance response of *D. officinale* under far-red light.

## lncRNAs involved in the signal transduction pathways responsible the effects of far-red light on the shade-avoidance response of *D. officinale*

Based on previous studies on the signal transduction pathways related to far-red light *(Kang et al., 2009*; *Zheng et al., 2013*; *Sharkhuu et al., 2014)*, this study established a regulatory network for the shade-avoidance response of *D. officinale* to far-red light (Fig. 8).

In the signal transduction pathway for the effects of far-red light in the shade-avoidance response of *D. officinale*, DE lncRNA MSTRG.38867.1 might act through the *PHYA* target gene, whereas the DE lncRNAs MSTRG.64192.1, MSTRG.64193.2 and MSTRG.64207.1 might act through the target gene *PHYB* (Fig. 8). Photosensitive pigments, which include Pr and Pfr phytochromes, play an important role in the shade-avoidance response. After absorbing one photon, Pr is converted to Pfr, and after absorbing another photon, it is

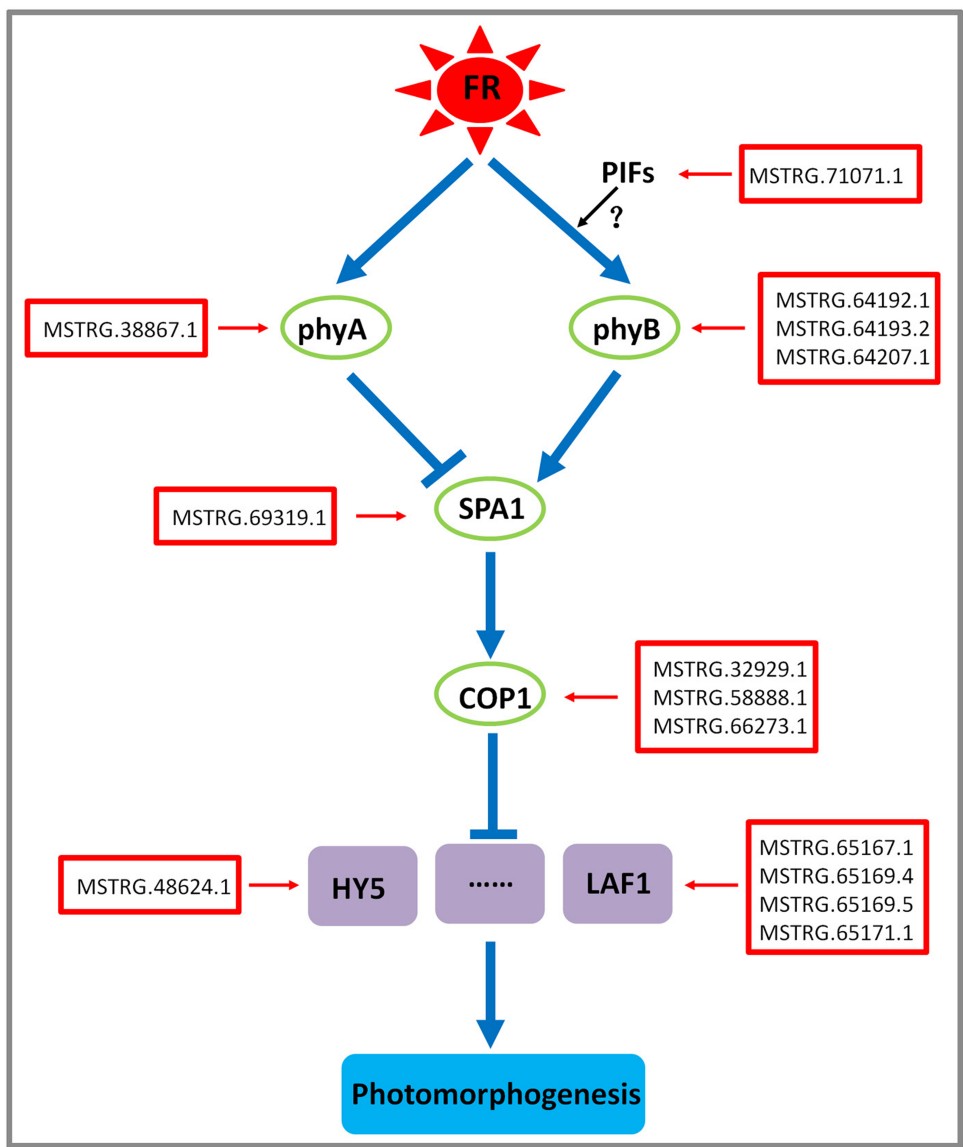

**Figure 8 Model depicting the regulatory lncRNA-mediated mechanisms of photomorphogenesis under far-red light.** FR, far-red light. The question mark indicates an uncertain relationship that requires further verification.

converted back to Pr (*Chen et al., 2018b*; *Chen et al., 2018a*). *PHYB* is a photoreceptor involved in most shade-avoidance reactions. Pfr is transported from the cytoplasm to the nucleus under shaded conditions such that it can more easily bind with transcription factors to exert its biological activity. *PHYB* can also sense a low R:FR ratio in shaded environments and can promote the conversion of Pfr to Pr (*Julia et al., 2010*). Studies of *A. thaliana* have shown that *PHYB* plays an important role in dark reversal and is a good sensor of light irradiance during early seed development (*Julia et al., 2010*). Because the light conversion response is slow under shaded conditions, the dark reversal of Pfr can

stably promote the conversion of Pfr to Pr. Although *PHYA* is not sensitive to the R:FR ratio, it is a sensitive sensor of changes in light irradiance during the shade-avoidance response, potentially because most photomorphological responses require the presence of Pfr in the nucleus (*Rausenberger et al., 2011*). In addition, *PHYA* is very important for the conversion of the dark phytochrome form to the light phytochrome form under shaded conditions, which results in a process called de-etiolation.[62]

The DE lncRNA MSTRG.71071.1 might participate in the signal transduction pathway underlying the shade-avoidance response of *D. officinale* to far-red light through the target gene PIF3 (Fig. 8). The main characteristic of PIFs is that they can interact with the Pfr form of the photosensitive pigment and thereby regulate the growth of stems as part of the shade-avoidance response (*Zhou, 2012*). Seven members of this subfamily, *PIF*1/*PIF*3-*LIKE*5 (*PIL*5), *PIF*3, *PIF*4, *PIF*5/*PIL*6, *PIF*6/*PIL*2, *PIF*7, and *PIF*8, interact directly with the conserved N-terminal sequence of *PHYB*, whereas two members, *PIF*1/*PIL*5 and *PIF*3, interact with *phyA* (*Zhou, 2012*; *Liu et al., 2011*). The shade-avoidance response under low R:FR conditions requires the participation of *PIF*3, *PIF*4, *PIF*5, and *PIF*7 (*Zhou, 2012*; *Liu et al., 2011*). For example, *PIF*3 functions mainly to control the morphological phenotype of plants and participates in some biochemical pathways involved in the light response process. Under high R:FR conditions, Pfr induces the phosphorylation of *PIF*7 and reduces the binding of *PIF*7 to the cotarget promoter to induce negative regulation. When plants are transferred to low R:FR conditions, *PIF*7 increases its binding to its target promoter and thereby exerts its effect (*Zhou, 2012*; *Liu et al., 2011*). These results indicate that the transition from a high to a low R:FR ratio under shaded conditions will reduce the active form of PHYB and thereby increase the activity of PIFs. An increased activity of PIFs will in turn promote the growth of plants during the shade-avoidance response (*Lau & Xing, 2010*).

The DE lncRNAs MSTRG.32929.1, MSTRG.58888.1, and MSTRG.66273.1 might act through the target gene *COP*1, and the DE lncRNA MSTRG.69319.1 might act through the target gene *SPA*1 and participate in the signal transduction pathway underlying the shade-avoidance response of *D. officinale* to far-red light (Fig. 8). COP1, a conserved RING finger E3 ubiquitin ligase, is involved in many biological processes, including plant growth and development and metabolic processes (*Pacín, Legris & Casal, 2013*). During emergence, plant seedlings undergo a transition from darkness to light. At this time, the expression of *cry*1, *PHYA*, and *PHYB* is induced by light, and COP1 slowly moves from the nucleus to the cytoplasm. SPA1 acts as a cofactor to regulate COP1 ubiquitin ligase activity (*Pacín, Legris & Casal, 2013*). In *Arabidopsis*, SPA proteins can self-associate with each other and form COP1/SPA heteropolymers, as reflected via various comparisons with COP1 (*Zhu et al., 2008*). These COP1/SPA complexes form part of the CULLIN 4-DAMAGED DNA-BINDING 1 ubiquitin E3 ligase complex (CUL4-DDB1$^{COP1/SPA}$) and mediate substrate recognition (*Lau & Xing, 2010*). Several positive regulators of the light response, including *HY*5, *HYH*, long after far-red light 1 (*LAF*1), and *HFR*1, are degraded by CUL4-DDB1$^{COP1/SPA}$-targeted 26S proteasomes (*Zhou et al., 2014*). Light-activated *PHYA* and *PHYB* downregulate the E3 ubiquitin ligase activity of CUL4-DDB1$^{COP1/SPA}$ and thereby stimulate the light response by stabilizing these positive regulators.

The DE lncRNA MSTRG.48624.1 might act through the target gene *HY*5, whereas the DE lncRNAs MSTRG.65167.1, MSTRG.65169.4, and MSTRG.65169.5 may act through the target gene *LAF*1 and participate in the signal transduction pathway underlying the shade-avoidance response of *D. officinale* to far-red light (Fig. 8). *HY*5 has an extremely wide spectrum of effects, including those caused by far-red, red, blue, and UV-B light (*Lockhart, 2014*). The results of this study showed that the HY5 protein level was positively correlated with the degree of photomorphogenesis (*Rodrigo et al., 2016*). Moreover, it has been found that the target genes of *HY*5 are related mainly to physiological processes such as photomorphogenesis, anthocyanin and chlorophyll synthesis, and lateral root development (*Sara et al., 2017*). *HFR*1 cannot directly interact with *PHYA* or *PHYB*, but HFR1/PIF3 dimers preferentially interact with *PHYA* and *PHYB* via Pfr (*Jang, Henrigues & Chua, 2013*). *HFR*1 regulates the expression of target genes by interacting with other basic helix-loop-helix (bHLH)-type transcription factor PIFs (*Hornitschek et al., 2009*). For example, *HFR*1 prevents a strong shade-avoidance response by forming a heterodimer that does not bind to DNA with *PIF*4 or *PIF*5. *PIF*4 and *PIF* 5 are two bHLH-type transcription factors that can directly regulate the expression of marker genes of the shade-avoidance response (*Hornitschek et al., 2009*). *HFR*1 and *PIF*1 interact to inhibit the binding of *PIF*1 and downstream target gene promoters and regulate seed development (*Shin et al., 2013*). *LAF*1 regulates the *PHYA* signalling pathway and also directly interacts with *HFR*1, and this interaction can inhibit the ubiquitination of *COP*1, stabilize *HFR*1 and *LAF*1, and promote the transduction of *PHYA* signals, which ultimately affects plant photomorphogenesis (*Jang, Henrigues & Chua, 2013*).

## CONCLUSION

This study provides the first demonstration of the effects of far-red light on lncRNAs involved in the shade-avoidance response of *D. officinale* through an RNA-seq analysis. We found that an appropriate proportion of far-red light can increase the leaf area, accelerate stem elongation, and thereby increases the production of *D. officinale*. Based on the transcriptomic, physiological and biochemical analyses, we revealed that lncRNAs involved in some metabolic pathways (i.e., flavonoid metabolism, alkaloid metabolism, carotenoid metabolism and polysaccharide metabolism) influence the effects of far-red light on the shade-avoidance response of *D. officinale*. The effect of far-red light on *D. officinale* might be closely related to cell signalling perception and conduction and $Ca^{2+}$ transduction. The GO and KEGG pathway enrichment analyses of the DE lncRNA targets indicated that hormone signal transduction and DNA methylation, among other functions, are likely to play a central coordinating role in the shade-avoidance response of *D. officinale* to far-red light. Based on the Cytoscape analysis and previous studies on far-red light-related signal transduction pathways, this study also found that some lncRNAs might participate in the far-red light signalling network through their target genes and thus regulate the shade-avoidance response of *D. officinale*. These findings provide new insights into the shade-avoidance response of *D. officinale* under far-red light and will be helpful for generating new ideas for the high-yield production of medicinal components of *D. officinale*.

## ACKNOWLEDGEMENTS

We thank American Journal Experts for editing the English text of a draft of this manuscript.

### Funding

This work was funded by the National Natural Science Foundation of China (31501802), the Natural Science Foundation of Fujian Province (2020J01377), the Education research project for young and middle-aged teachers in Fujian (JAT190696), the Sanming University Scientific Research Foundation for High-level Talent (18YG01, 18YG02, 19YG06), and the 2019 and 2020 Special Commissioner of Science and Technology of Fujian Province. The funders had no role in study design, data collection and analysis, decision to publish, or preparation of the manuscript.

### Grant Disclosures

The following grant information was disclosed by the authors:
National Natural Science Foundation of China:  31501802.
Natural Science Foundation of Fujian Province:  2020J01377.
Education research project for young and middle-aged teachers in Fujian: JAT190696.
Sanming University Scientific Research Foundation for High-level Talent:  18YG01, 18YG02,  19YG06.
2019 and 2020 Special Commissioner of Science and Technology of Fujian Province.

### Competing Interests

The authors declare there are no competing interests.

### Author Contributions

- Hansheng Li and Wei Ye performed the experiments, analyzed the data, authored or reviewed drafts of the paper, and approved the final draft.
- Yaqian Wang, Xiaohui Chen and Yan Fang analyzed the data, authored or reviewed drafts of the paper, and approved the final draft.
- Gang Sun conceived and designed the experiments, analyzed the data, prepared figures and/or tables, authored or reviewed drafts of the paper, and approved the final draft.

### DNA Deposition

The following information was supplied regarding the deposition of DNA sequences:
All sequencing data are available in the National Center for Biotechnology Information (NCBI) Sequence Read Archive: PRJNA638348.

### Data Availability

Raw data are available as a Supplementary File.

## Supplemental Information

Supplemental information for this article can be found online at http://dx.doi.org/10.7717/peerj.10769#supplemental-information.

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
