# Peer review of "RNA sequencing-based exploration of the effects of far-red light on lncRNAs involved in the shade-avoidance response of D. officinale"

_PeerJ, doi:10.7717/peerj.10769_

## Round 0.1 · original submission · Major Revisions

We have received the reports from our advisors on your manuscript "RNA sequencing-based exploration of the effects of far-red light on lncRNAs involved in the shade-avoidance response of D. officinale".
I have some suggestions:

1.The language in this manuscript requires polishing.

2.A uniform format is needed for the “References” part.

3.Additional data are required for the qPCR assays reported. You must show that the assay for each gene was adequately validated. Please refer to the MIQE Guidelines for a detailed discussion of the specific parameters that should be included in the manuscript:

Bustin SA, et al. The MIQE guidelines: minimum information for publication of quantitative real-time PCR experiments. Clin Chem. 2009 Apr;55(4):611-22. doi: 10.1373/clinchem.2008.112797. Epub 2009
Feb 26. PubMed PMID: 19246619.

The minimum acceptable parameters, such as assessment of RNA quality, efficiency/reproducibility of the reverse transcription assay, the demonstration that you are generating a single PCR product of the expected size, as well as key parameters of the qRT-PCR reaction such as PCR efficiency, linear dynamic range (particularly with respect to the input cDNA concentrations you are using), LOD and some estimation of the assay precision must be included. Since some of the results supporting your conclusions are based on qRT-PCR, it is essential that you demonstrate to the reviewers that you have a robust assay. I would suggest providing these data in the supplemental data with all the essential information for each assay combined in a single figure/table.

4.The content of sequencing needs to be described in detail in the method.


Below, please find the comments for your perusal.

Reviewer 1 ·

Basic reporting

No comment

Experimental design

Your introduction on Materials & Methods needs more detail, especially on Plant materials. How often the plantlets were subcultured was not introduced in this paper?I don’t think they were stayed in one bottle for 120 days. And I suggest that you give us a picture of the seedling used in the experiment before the treatments.

Validity of the findings

No comment

Additional comments

In the manuscript ,the authors used high-throughput sequencing technology to identify putative lncRNAs and investigated their expression profiles in Dendrobium under different illumination patterns. By comparing and analyzing the sequence data of the treatment group and the control group, the specific lncRNAs were studied for their involvement in the shade-avoidance response of D. officinale caused by far-red light. While, there are still some points , which need to be improved and corrected in the manuscript.
There are other issues that need to be revised:

1) The species and genus names of the experimental materials are confused in the manuscript. Please unify D. candidum and D. officinale.

2) Fig 1, I suggest that you give us a picture of all the seedling(all the different treatment) used in the experiment. There should be a scale as the reference in Fig 1.


3)Line 113 “200 μmol/m2/s” is wrong, please correct it. I think it should be 200 µmol·m-2·s-1.

4) The author made a serious mistake in the introduction of experimental materials in “RNA sequencing (RNA-seq) library construction and sequencing”. Line 124, “According to the phenotypic differences of D. officinale, three treatments-CK, FR1 and experimental group 5”, while we just can just see the results CK, FR1 and FR4. The authors are very careless.

5)In the section INTRODUCTION, the author mentioned that lncRNA has been reported in many plants, but the number of lncRNA identified in this study is far less than that of other species. What is the reason? Is the test data reliable?


6)The logic was weaker in RESULT. Please give us more evidence for the goal about the experiment. In the manuscript, after identified the lncRNA by RNA-seq, qRT-PCR and bioinformatics prediction are the main means for the author to obtain the experimental results. I think you need more evidence to support your results and conclusion.

7) Fig. 5 is very obscure.

Reviewer 2 ·

Basic reporting

no comment

Experimental design

no comment

Validity of the findings

no comment

Additional comments

The present manuscript describes the analysis of lncRNA differential expression under different red:far-red light conditions in D. officinale.
Even though the article is well orientated I believe that it should be restructured and some extra analysis included. Some comments and suggestions are included below:

Line 175-180 Results are delivered without any context and statistical significances of differences in plant height, stem length, etc are not reported. The number of biological replications is not clear. I suggest the authors include this information.

Line 195 There is little context to support the study of lncRNA over, for example, gene expression. I recommend the authors to briefly introduce their hypothesis.

The article will benefit from the inclusion of a PC analysis on the structure of the sequencing of the biological replications.

Line 205. I wonder why authors avoid differentially expressed genes. What proportion of all the reads mapped to the genome are mapped in lncRNA? What proportion of genes is affected? Is there any of this lncRNA associated with differentially expressed genes? Can you summarize lncRNAs and putative gene target? Are there any retroelements between these lncRNAs?

Line 290. The authors should consider restructuring this section to help the reader to follow the text. The mere description of a big number of genes and pathways is very hard to follow. Positive or negative correlations should be supported by statistical differences.

The Discussion section goes deep in the analysis of important genes related to these responses but does not mention if these were differentially regulated in the present study.

The inclusion of gene pathways and the annotation of new roles in those should be avoided. The authors should mention the limitations of the current study in assigning roles and functions with only differential expression data.

Reviewer 3 ·

Basic reporting

The current study is well-organized and discussed with good scientific quality. However, the authors should incorporate proposed comments and issues to improve the presentation of their work.

Experimental design

Check the below comments.

Validity of the findings

Check the below comments.

Additional comments

Material and methods
 Line 127 and 132, 12 h/d, 6 g/L. Please use standard style throughout the text, i.e., 12 h d-1, 6 g L-1. Check the entire manuscript and fix this issue.
 Line 177, add a full stop at the end of the sentence (Fan et al. (2017).
 Line 182-187, authors must include the qRT-PCR ingredients profile and volume.
 Provide the developer name and location of SPSS and GraphPad Prism.

Results and discussion
 Please use standard style throughout the text, i.e., mg g-1, g L-1. Check the text, figures, and tables, and fix this issue. E.g., Figure 1. Line 207, 208, 210,
 In table S2-S7, remove dot between (mg·g -1 DW). It should be mg g-1. Further, replace u with the micro symbol in table S6 and S7. Also, check other tables and text for this error.
 Line 249-266, mention the GO IDs along with terms.
 The caption of figure 4D “The green color indicates that the comparison contains this pathway, and the yellow color indicates that the comparison does not contain this pathway”. Please note that there is no yellow color in the figure. It should be green. Please check and modify the caption or color in the figure accordingly.
 Line 304, add the Cytoscape version used in this study.
 In figure 5, authors should add an expression heatmap scale for both figures, indicating the highest and lowest values, to make it self-explanatory.
 Line 303-326, the authors constructed an interaction network of lncRNA-mRNA, which is based on the cis roles of lncRNAs involved in their action on neighboring target genes. Further, the authors searched the coding genes that were 10 kb/100 kb upstream/downstream of lncRNAs and then subsequently analyzed their function. However, I cannot see the validity of this interaction network. The authors must have considered a correlation analysis like the Pearson coefficient correlation analysis to analyze the strength of lncRNA-mRNA interaction in terms of positive or negative interaction/correlation.
 Line 306-326, please cite suitable supporting data, i.e., table S8 or S9 to support your results.
 Line 406, what is the meaning of “29”? Is there was any reference? Please check and add the appropriate reference.
 I suggest the authors cite suitable figure or table in the discussion part as well.
 In figure 10, please explain the abbreviations in the caption.
 Define the abbreviations the first time they appeared in the text. Check the entire document and fix this error, if any.
 Authors must include a separate section of “conclusion”.

Please make sure all the cited references are given in the list and vice versa. Further, prepared references according to the authors guidelines.

---

## Round 0.2 · Minor Revisions

Dear Dr. Li,

Thank you for your submission to PeerJ.

It is my opinion as the Academic Editor for your article - RNA sequencing-based exploration of the effects of far-red light on lncRNAs involved in the shade-avoidance response of D. officinale - that it requires a number of Minor Revisions.

My suggested changes and reviewer comments are shown below and on your article 'Overview' screen.

Please address these changes and resubmit. Although not a hard deadline please try to submit your revision within the next 40 days.

Reviewer 1 ·

Basic reporting

No comments

Experimental design

No comments

Validity of the findings

No comments

Additional comments

1. The authors detected the metabolite contents in D. officinale under different light treatments, such as flavonoid, alkaloid and carotenoid contents . But why not detect the contents of polysaccharide ? As we known, polysaccharide is the main active component of D. officinale.
2. The title of Figure 7 is not appropriate, please revise it. qPCR is the abbreviation of quantitative real-time PCR, so you need delete “quantitative”. And the Y axis in the Fig.7 should be relative expression level.

Reviewer 2 ·

Basic reporting

no comment

Experimental design

no comment

Validity of the findings

no comment

Additional comments

The authors addressed the comments and suggestions made in the previous review.

Reviewer 3 ·

Basic reporting

Dear Authors,
Thank you for revising the MS according to the proposed comments and suggestions. Notably, MS has been improved. I have only one suggestion before acceptance.
Please define PIF in the introduction on the first appearance. phytochrome- interacting bHLH factor (PIF).

Experimental design

No comments.

Validity of the findings

No comments.

Additional comments

No comments.

---

## Round 0.3 · Minor Revisions

Dear Dr. Li,

Your article upon review requires a few more items addressed. The manuscript reads well; however, is missing critical data to allow the reader to evaluate the findings. It is mentioned that a reference genome is available, but there are no gene_IDs or even sequence data to tie the annotations created to actual sequence data. In lines 95-97 it is highlighted that works in other organisms have yet to investigated lncRNA light-regulation, yet a simple search identified similar studies in all the organisms mentioned; the end of the sentence should be instead “… investigated in Dendrobium.”. Latest suggestions are that at least 3 replications be available for each experiment performed; this does appear to be the case from the PRJNA638348 SRA data. It’s not clear if sequence data is available from the reference genome mentioned in line 140; it should be, and coordinates for available data would be valuable; otherwise the set of deferentially expressed sequences in FASTA format from the transcriptome should be available with appropriate annotations in a table, or supplement (not the statistics), but the actual 3,086 new genes, 2,125 DE genes, 3,770 lncRNAs, and 136 DE lncRNAs; otherwise, there is little to learn from the presentation. It would be valuable to know which of the annotations from Table 3 are attached to the gene sequences. In line 281 there is mention of an annotation related to diabetes, yet nothing else is discussed if this has any relevance, or that some comparison can be made to plant systems to make this valuable information. I will classify this as requiring additional revision until some of the observations can be addressed. It is in general interesting, but provides nothing for the reader to evaluate it data-wise.

---

## Round 0.4 · Minor Revisions

It appears the suggested revisions have been addressed in the rebuttal and resolved as best as possible. Additional files were included in the supplemental data provided; however I still do not see any mention of the reference genome mention regarding version used. Yes there a link to a publication, but it is important to help guide the reader to the data rather than establish a cryptic hunt for the data. The assembly data was mentioned in the rebuttal letter, but there were no changes to the manuscript to reflect the notes mentioned. Where in the v3 version does the ASM160598v2 version mentioned? I did not see a pointer to the NCBI assembly data; I did see the reference to the raw data, but not the assembly. Please add some guidance to the manuscript. A manuscript should be informative and help lead the reader in a fashion to be in a position to validate the presented work. The data is hidden in a fashion which is a tedious effort for the reader to follow. This is almost ready but still requires revision to clear up what data is available and were it is.

---

## Round 0.5 · accepted · Accept

The added link to the manuscript proves the needed link to data resources. I feel the manuscript is now in shape to move forward; I will provide a recommendation to accept the manuscript in its current form. Thank you for your patience during these final stages of the review.